# Global scenarios of resource and emission savings from material efficiency in residential buildings and cars

Stefan Pauliuk [1✉], Niko Heeren [2,3], Peter Berrill[2], Tomer Fishman [4], Andrea Nistad[3], Qingshi Tu[2,5], Paul Wolfram [2] & Edgar G. Hertwich [2,3✉]

Material production accounts for a quarter of global greenhouse gas (GHG) emissions. Resource-efficiency and circular-economy strategies, both industry and demand-focused, promise emission reductions through reducing material use, but detailed assessments of their GHG reduction potential are lacking. We present a global-scale analysis of material efficiency for passenger vehicles and residential buildings. We estimate future changes in material flows and energy use due to increased yields, light design, material substitution, extended service life, and increased service efficiency, reuse, and recycling. Together, these strategies can reduce cumulative global GHG emissions until 2050 by 20–52 Gt $CO_2$-eq (residential buildings) and 13–26 Gt $CO_2$e-eq (passenger vehicles), depending on policy assumptions. Next to energy efficiency and low-carbon energy supply, material efficiency is the third pillar of deep decarbonization for these sectors. For residential buildings, wood construction and reduced floorspace show the highest potential. For passenger vehicles, it is ride sharing and car sharing.

[1] Industrial Ecology Group, Faculty of Environment and Natural Resources, University of Freiburg, Freiburg, Germany. [2] Center for Industrial Ecology, School of the Environment, Yale University, New Haven, CT, USA. [3] Industrial Ecology Program, Norwegian University of Science and Technology (NTNU), Trondheim, Norway. [4] School of Sustainability, Interdisciplinary Center (IDC) Herzliya, Herzliya, Israel. [5] Department of Wood Science, University of British Columbia, Vancouver, Canada. ✉email: stefan.pauliuk@indecol.uni-freiburg.de; edgar.hertwich@ntnu.no

Achieving the Paris Agreement goal of limiting global warming to well below 2 °C requires a rapid decarbonization of the economy, which, according to most climate-economic models, can only be done with the use of costly carbon-removal technologies[1,2]. The decarbonization of industry and material production, in particular, requires technological and organizational change and large investments into new energy infrastructure and factories[3]. Greenhouse gas (GHG) emissions from material production have risen from 5 Gt $CO_2$-equivalents ($CO_2$-eq) in 1995 to 11.5 Gt in 2015[4] and represent about 23% of global GHG emissions. The median remaining lifetime of existing production facilities for cement and steel stretches to 2045, causing substantial lock-ins that impede decarbonization efforts in this sector[5]. Decarbonizing material production requires further technological development[3,5–7] and will compete with other applications of low carbon energy, including electric transportation and low-temperature heat[8]. Given the anticipated slow pace of decarbonizing material production, the reduction of material demand through (i) more efficient use of materials at all stages of the material cycle[9] and (ii) the decoupling of services, such as mobility, from the number of material-intensive products, such as vehicles[10], may result in more immediate emission reductions. Governments are hence assessing or implementing policy frameworks to reduce material demand[11], variously referred to as material efficiency (ME)[12,13], resource efficiency, a sound material-cycle society, sustainable material management, or the circular economy[14]. ME strategies in fabrication and waste management aim at prolonging the technical lifetime of engineering materials; they are also termed value-retention strategies and form the core of the circular-economy vision[15].

To put ME into perspective and inform policy-making, model-based assessments are needed to quantify the potential system-wide impact of different ME strategies, in particular on material production, energy and raw material demand, and GHG emissions. Models also inform about interactions among ME strategies and assess their potential under different energy system and socioeconomic futures. Single materials, products, and sectors have been studied extensively[12,16–18], often using life cycle assessment, but there is a gap between these detailed assessments and the more aggregate representation of the industrial system in the climate–energy–economic models used for integrated assessment[2,19]. The former tend to overlook the implications of large-scale strategy rollout on material cycles and the changing technology landscape outside of material processing, whereas the latter typically lack a representation of technological detail and mass balances to capture engineering innovations that reduce material demand or the availability of materials for recycling[20]. Attempts to bridge that gap are emerging. Industrial ecology research has provided a number of scenario analyses for the future demand and supply of specific materials and metals[21–23]. In the integrated assessment modelling community, van Ruijven et al.[24] developed a gross domestic product (GDP)-driven high-resolution scenario model for steel and cement demand and production. However, these material demand and supply scenarios have not been linked to service provision and do not include a detailed depiction of ME. The multi-sector low energy demand (LED) scenario[25] contains a detailed depiction of end-use energy-related demand-side mitigation strategies. It also includes the mitigation potential of ME but in a very aggregated manner, by applying a demand-side 'dematerialization multiplier' and a supply-side 'material efficiency' term. These simplifications limit the ability of these models to accurately quantify the effect of ME on material cycles and related energy use/GHG, and thus to identify the most promising strategies. A detailed review of the grey literature assessing ME is contained in Section 1 of the Supplementary Material.

The current assessment focusses on potential GHG emission reductions and impacts on material stocks, and flows of material-efficient residential buildings and passenger vehicles. The production of materials used in residential buildings and cars in 2015 was estimated to have caused GHG emissions of 2.4 and 0.8 Gt $CO_2$-eq, respectively[4,11]. Whereas, in 2018, the operational energy use of these products caused emissions of 6.0 and 7.5 Gt $CO_2$-eq, respectively, and accounted for 21% and 18 % of total final energy consumption (see Section 2 of the Supplementary Material).

Here we estimate the global GHG mitigation potential of a broad and ambitious rollout of ME. GHG emission savings until 2050 are 20–52 Gt $CO_2$-eq (residential buildings) and 13–26 Gt $CO_2$e-eq (passenger vehicles), depending on the climate policy scenario. After full implementation of ME in 2040, GHG savings of remaining emissions of 1/3–2/3 are observed in both sectors and all regions studied, which demonstrates that ME can be key to deep decarbonization or climate neutrality. Demand for climate-impactful materials such as steel and cement declines substantially.

## Results and discussion

Our analysis covers the resource and GHG impact of ME in the residential-building and passenger-vehicle sectors, covering the entire world comprising 20 countries/regions, grouped into the Global North (OECD (Organisation for Economic Co-operation and Development), former USSR countries, China) and the Global South (low- and medium-income countries in Asia, Africa, and the Americas). The ten ME strategies assessed include the following: supply-side measures (higher yields in fabrication and scrap recovery, reuse of fabrication scrap, and product lightweighting through better design/downsizing or material substitution) and demand-side measures (reuse of products and product-lifetime extension (longer use), sufficiency-related measures including more efficient use of cars via car sharing and ride sharing, and more intense and efficient use of dwelling space resulting in less floorspace per person). When implemented in a given scenario, the full technical potential for each ME strategy is assumed to be realized by 2040. The assessment considers three socioeconomic scenarios, an LED scenario[25], and two of the shared socioeconomic pathways[26], SSP1 and SSP2, representing low and intermediate socioeconomic challenges related to climate-change adaptation and mitigation, respectively. Two policy scenarios are considered for each SSP, one with no new climate policy after 2020 and one for decarbonizing the energy supply and widespread electrification to limit the average temperature rise to 2 °C (i.e., the representative concentration pathway of 2.6 W/m$^2$ additional forcing, RCP2.6)[27]. The model captures the production, demand, use, and recycling of six major climate-relevant materials (aluminium, cement, copper, plastics, steel, and wood) for the period 2016–2060 (results reported for/by 2050), starting from 2015 as the last year with complete empirical data.

On the basis of the LED and SSP scenario storylines[28], we developed parameter values using a combination of data-driven extensions of historical data, literature studies, and expert consensus approaches, similar to the development of the SSP scenarios framework itself. These parameters include future service level (passenger-km delivered by cars, residential floor area utilized) and the share of the different drive and building technologies used. Future service levels were subject to several rounds of consensus building and refinement, documented in detail in an accompanying study[29,30].

Whereas the LED values were only slightly modified when breaking them down from the Global North/South split to individual countries, the SSP2 values continue (Global North) or

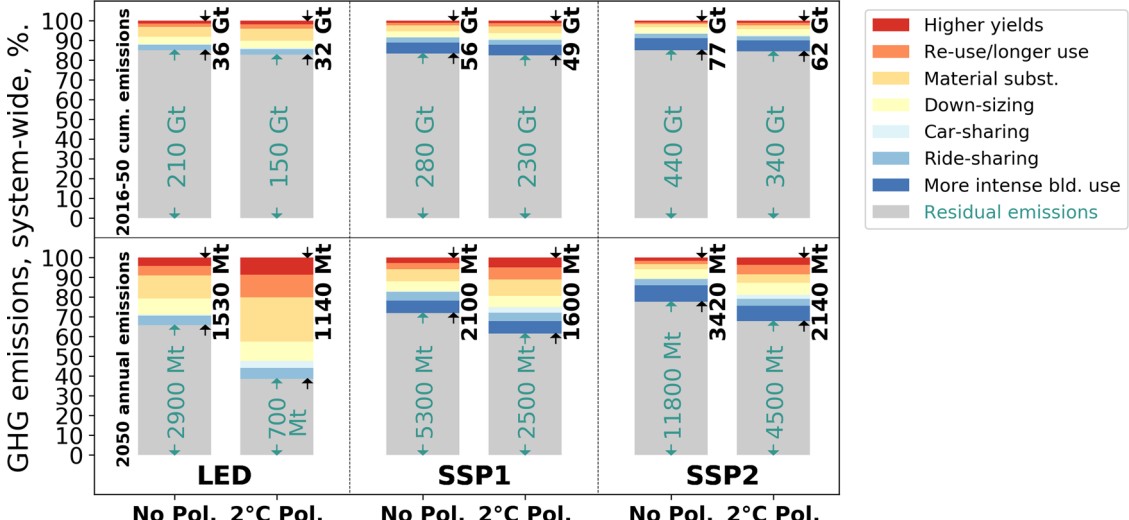

**Fig. 1 Total global cumulative (top) and annual 2050 (bottom) greenhouse gas (GHG) emission reductions of the technical potential of ten industrial and demand-side material efficiency (ME) strategies.** Results are shown for three socioeconomic (low energy demand (LED) and the shared socioeconomic pathways (SSP1 and SSP2)), and climate policy (No Pol. and 2 °C Pol., see text) scenarios and ME strategy for the passenger-vehicle and residential-building sectors combined. The absolute values in the plot are in megatons (Mt) or gigatons (Gt) of $CO_2$-eq. See the 'Methods' section for an overview of the different ME strategies implemented.

converge to (Global South) service levels currently experienced by citizens in the Global North. The SSP1 values typically describe a compromise between the LED and SSP2 trends (see the 'Methods' section). Except for extrapolations of service levels in the SSP2 scenario, GDP is not used as a model driver; the scenarios are GDP agnostic[31].

**Global GHG emission savings of ME**. The different ME strategies combined can reduce cumulative global GHG emissions of the period 2016–2050 by 32–77 Gt (13–18% of the total), depending on socioeconomic development and climate policy (Fig. 1, top row, see the 'Methods' section for scenario settings). All examined strategies show a visible contribution (numerical values reported in the data supplement). For the LED scenario, where in-use stocks are already used very intensively (low floor-space per capita), material substitution, reuse, and longer use are the ME strategies with the largest GHG reduction potential. For SSP1 and SSP2, more intense building use and material substitution show the largest contribution, followed by downsizing, reuse, and longer use. The ME strategy car sharing shows much larger contributions in the 2 °C policy mix. The reason for that is that this scenario has a higher share of electric vehicles, which are introduced faster, because car sharing reduces the vehicle fleet size but increases the average annual kilometrage, thus shortening vehicle lifetime, which increases the turnover of the fleet.

Once fully implemented, ME strategies can lead to large reductions of annual global GHG emissions. In 2050, annual savings can be between 22% and 61%, depending on ME stringency, energy-sector decarbonization, and anticipated growth in services (Fig. 1, bottom row). ME can make an important contribution to keeping anthropogenic GHG emissions within the remaining emission budget available for limiting global warming below 2 °C. Therefore, ME can reduce the risk and magnitude of emission overshoot and the need for negative emission technologies. Annual emission cuts from ME in 2050 are smaller in absolute terms but more important (as a share of the total) in the 2 °C scenario with a low-carbon energy supply compared to the case with no additional policy to drive further decarbonization. In a low-carbon energy future, ME-induced reductions of the difficult-to-mitigate GHG emissions in material

production have a relatively high impact in the system's GHG balance compared to energy-supply impacts. On the other hand, ME strategies will be crucial for delivering substantial GHG emission reductions in a future with resource intensive socio-economic development and without stringent climate policy Fig. 1, SSP2 No Pol.).

**GHG emission savings by sector and region**. The considered supply and demand-side ME strategies lead to a reduction of the use phase and production/construction-related GHG emissions of the vehicle and building sectors across all world regions and climate policy scenarios (Fig. 2). The vehicle sector in high-income countries/regions experiences a moderate decline in GHG emissions if no additional climate policies are issued and sub-stantial decline with stringent climate policy (especially an elec-trification of the fleet, combined with low-carbon electricity supply). Countries in the Global South are poised for further growth in sectoral emissions, but stringent climate policy and ME can mitigate emission growth to enable an earlier and lower peak (around 2035 instead of 2050). Emission reductions are more pronounced for residential buildings, as the energy mix is already relatively electrified to begin with, and emissions fall rapidly due to the decarbonizing electricity supply. Stock turnover and ret-rofits such as better insulation and heat-recovery ventilation further improve efficiency, and the replacement of oil and gas furnaces with heat pumps further increases electrification. In industrialized countries, emissions are set to decline even under current policies.

Using wood from sustainable forestry as long-lived construc-tion material where available[32,33] can lead to additional emission savings of 1–2 Gt/yr, depending on how much of it is used. In some regions of the Global South, the regrowth of forest in response to sustainably harvested timber for residential buildings can almost offset the emissions from the production of other construction materials by around 2050 (values close to zero in Fig. 2). Next to wood use in buildings, a development towards more intense use of buildings (modelled as lower average floorspace per capita) is a highly effective mitigation strategy that combines sufficiency with large energy and material savings in all countries and regions.

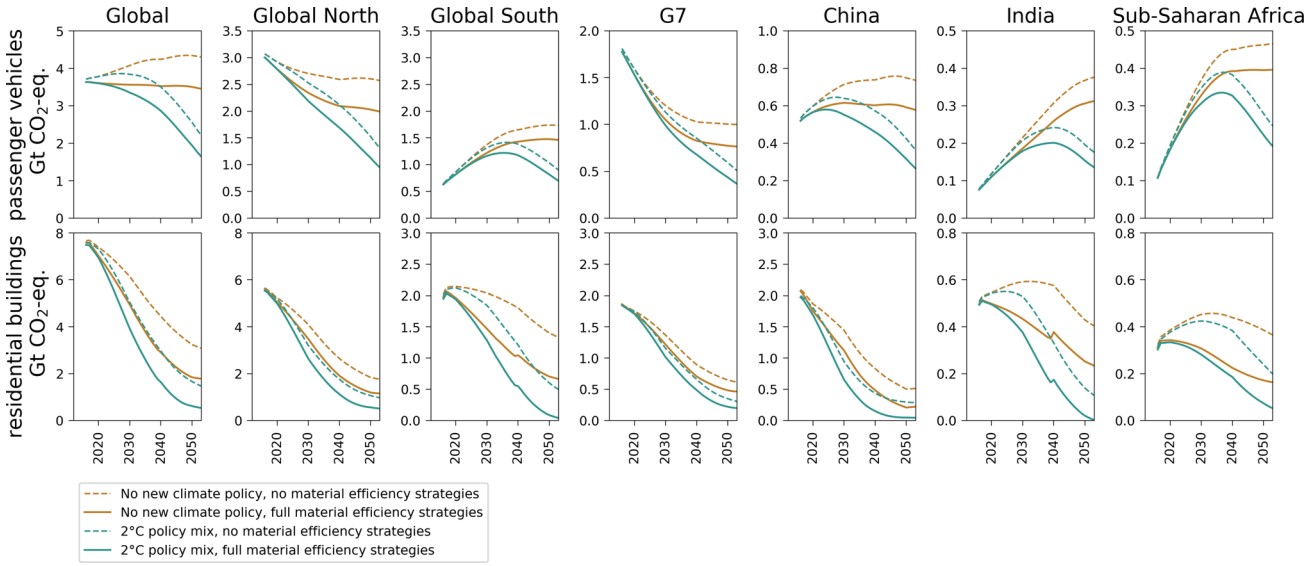

**Fig. 2 Greenhouse gas (GHG) emission scenarios (in gigatons (Gt) of CO₂-eq/yr) by region and time.** Results are shown for passenger vehicles (top row) and residential buildings (bottom row) for the SSP1 shared socioeconomic pathway (easy adaptation and mitigation) and two climate policy scenarios, with no material efficiency (ME) strategies considered and the full spectrum of ten strategies considered. See Section 5 of the Supplementary Material for scenario results for the other scenarios and for all 20 model regions. G7 refers to the Group of Seven countries.

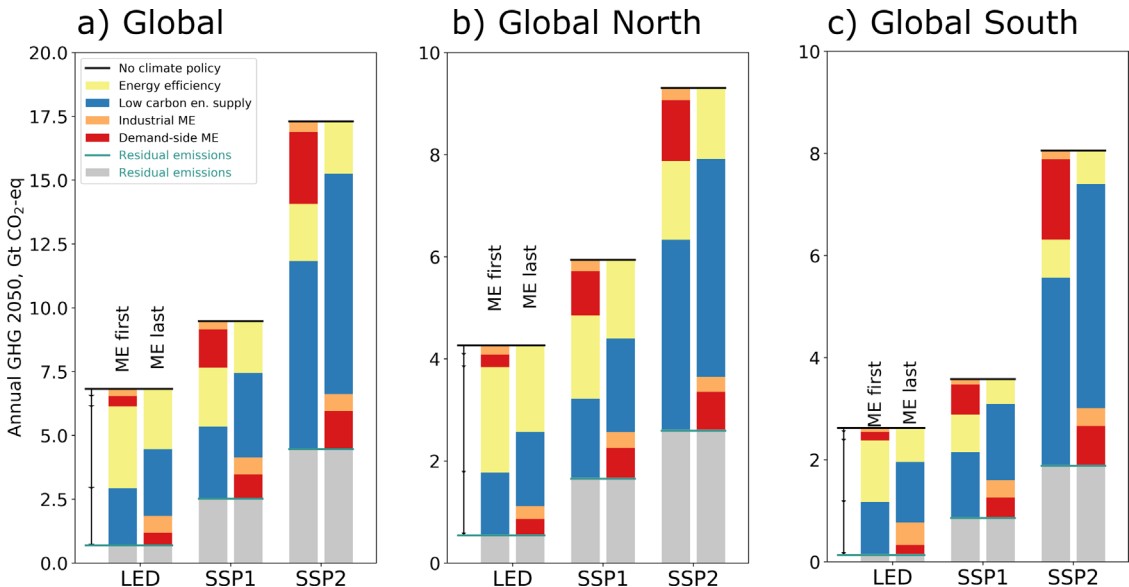

**Fig. 3 Emissions mitigation potential of material efficiency, energy efficiency, and low carbon energy supply combined.** Breakdown of total greenhouse gas (GHG) emission savings from baseline with no new climate policy (black horizontal line on top of bars) into end-use energy efficiency, energy supply (en. supply), industrial and demand-side material efficiency (ME), and for passenger vehicles and residential buildings combined, at the global level (**a**), the Global North (**b**), and the Global South (**c**). Three socioeconomic scenarios are shown: low energy demand (LED) and the shared socioeconomic pathways (SSP1 and SSP2). For the left bar in each scenario, ME was implemented first, before adding energy efficiency and low-carbon energy supply. For the right bar, ME was applied in addition to energy efficiency and low-carbon energy supply. The two red-coloured segments cover the ten ME strategies. Industrial ME includes recovery ratios for recycling, fabrication yield and scrap diversion, reuse, and material choice. Demand-side ME includes product light-weighting/downsizing, lifetime extension, car sharing, ride sharing, and more intense use of buildings. GHG emissions are reported in gigatons (Gt) of CO₂-eq.

**ME as third pillar of deep decarbonization**. The contextual analysis (Fig. 3) shows that due to the dominance of energy-related GHG in the global emissions budget, energy efficiency and a low-carbon energy supply are key to curbing global warming. However, even with these measures fully implemented in the two sectors studied, 2050 residual emissions are still substantial (e.g., 4.1 Gt for SSP1) and—if no other measures are taken—are likely to require compensation by negative emissions technologies to achieve carbon neutrality mid-century[2]. Therein lies the main contribution of ME

to GHG emission reduction. ME offers additional emission reduction opportunities that can help bridge the gap between a 2 °C and 1.5 °C future, as evident in Figs. 2 and 3. ME strategies are also less subject to concerns of feasibility, scalability, burden shifting, and rate of deployment that are associated with negative emission technologies[2,34,35].

The model-estimated contribution of mitigation strategies to overall emission reduction depends on their sequencing. In the bars on the right side of each scenario in Fig. 3, energy efficiency

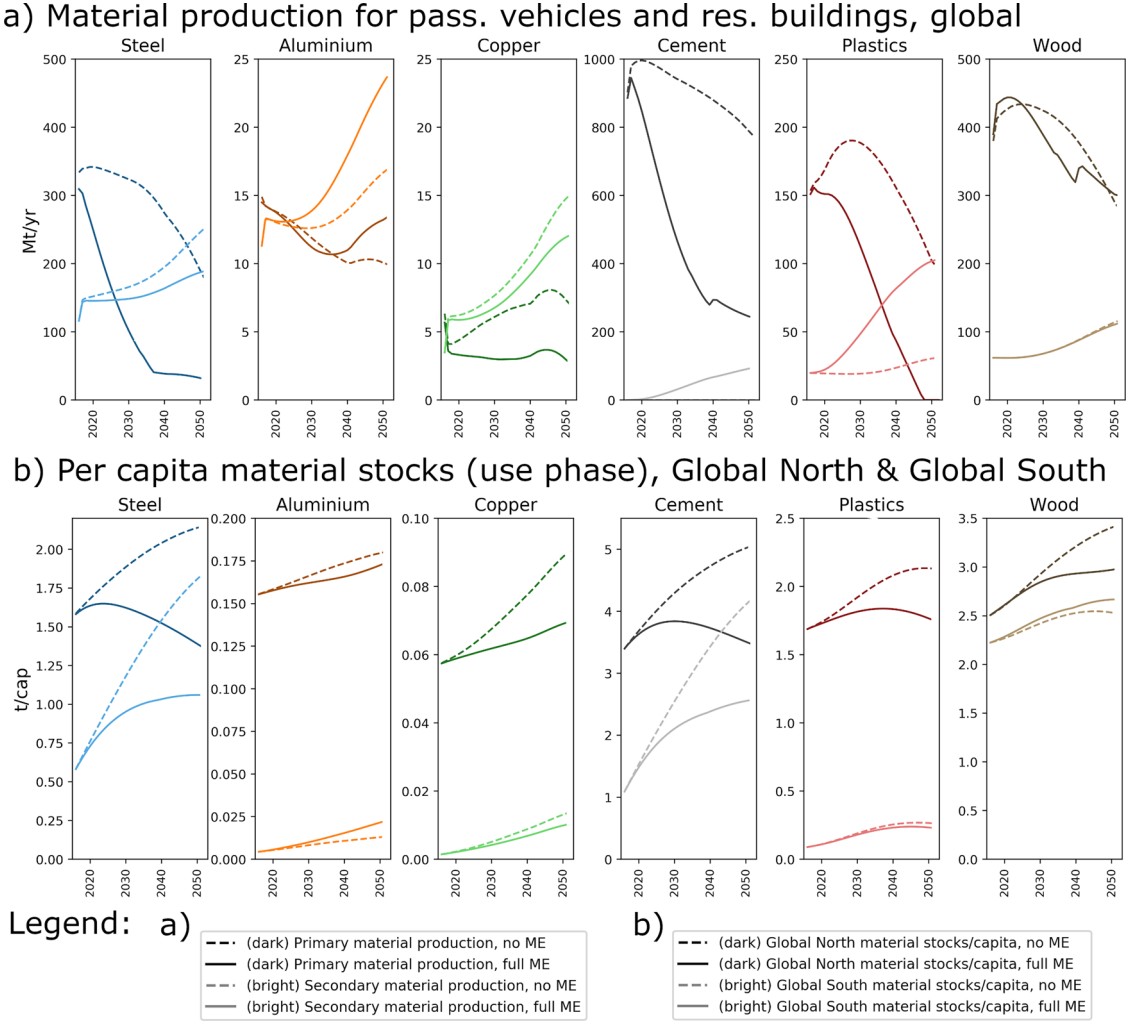

**Fig. 4 Impact of material efficiency on material production. a** Global material production 2016–2050 (primary = from virgin resources, secondary = from post-consumer scrap) for six major materials in SSP1 and a 2 °C policy mix, for passenger vehicles and residential buildings, and for scenarios with no and those with full material efficiency (ME). **b** Per-capita in-use stocks (2016–2050) of materials in passenger vehicles and residential buildings, Global North, and Global South average. The unit for part **a** is megatons per year (Mt/yr) and for part **b** it is ton per capita (t/cap).

and low-carbon energy supply are introduced first, and ME strategies are then applied on an already decarbonized system, yielding higher savings from decarbonization and lower savings from ME than if the sequence was reversed (left side bars). These two alternative sequences show that the impact of ME is larger for SSP1 and SSP2 in a world with high-carbon energy supply, which is a direct consequence of the carbon intensity of material production and of the use phase-related energy savings mediated by ME. The situation is different for LED, where the GHG savings potential of ME after implementing energy efficiency and low-carbon energy supply is larger than for the opposite sequence, especially for the Global South. The main reason for that effect is that material substitution, which dominates ME GHG savings in LED (see also Fig. 2), becomes much more effective once aluminium production is decarbonized (vehicle steel substitute), which is the case in the right bar but not in the left bar. After seizing the energy efficiency (green) and energy supply transformation (blue) potentials, the share of remaining global emissions reduced through ME is smaller in SSP2 (32%) and SSP1 (39%) than in LED (62%), because ME strategies are applied more gradually and to less ambitious end targets in SSP2 and SSP1, reflecting the storylines of those scenarios.

**Material-cycle impacts of ME.** In 2016, material demand of the two sectors studied absorbed about 430 Mt of steel and 900 Mt of cement (Fig. 4), corresponding to roughly 26% and 22% of the global steel and cement production, respectively. The impact of ME on primary and secondary material production at the global level is substantial, because of massive reductions in demand for primary (produced from virgin natural resources) steel, cement, copper, and plastics (Fig. 4a). In the Global North, steel and cement demand drop, because demand for new residential floorspace plummets, as more intense use leads to a re-purposing and contraction of the existing stock rather than an expansion of living space. In the ME scenarios, excess steel scrap from demolished buildings and de-registered vehicles is recycled for use in the Global South, where it bolsters growth of in-use stocks and helps raise living standards and urbanization, in particular (Fig. 4b). Demand for new plastics drops for two reasons. First, a lower stock growth due to more intense use (same as for steel and cement). Second, a substantial increase of the end-of-life recycling rate of plastics from today's 18% on average[36] to up to 70%, factoring in better product design (eco-design and design for dismantling) and the need to dilute recycled plastics with virgin material to maintain material quality. In addition to saving energy

and GHGs, reduced primary production will also lower industrial use of mineral resources, land, and water, thus yielding multiple co-benefits, which have yet to be quantified[22]. The material production volumes (Fig. 4) only include the demand and scrap supply of the two sectors studied, and the ratio between primary and secondary production reflects the sector-specific material stock dynamics and not the global total for the individual metals. Copper is an interesting example here, as its global average recycled content is below 40%, mainly due to large losses in electronics[37,38], but for vehicles and buildings, scrap recovery rates are high and the recycled content in the material supply for these two sectors can be 60% and higher.

Implementing ME at full technical scale does not mean that we use less of each material. There will rather be a higher demand for substitution materials such as aluminium and, temporarily, wood. Copper demand grows mainly because of the electrification of the passenger-vehicle fleet. The vehicle-material substitution scenarios are based on aluminium, because a large-scale supply of low-carbon aluminium requires only a change in the electricity source and is hence expected to arrive earlier than low-carbon steel, which requires entirely new facilities and production processes that are not expected to reach broad rollout before around 2035.

For wood, the increased demand from timber-based buildings is compensated for by the overall reductions from other ME strategies and more intense building use, in particular. The same trade-off applies to secondary materials, where overall throughput reduction from—among others—product light-weighting and lifetime extension is larger than the increase from higher recycling ratios for steel, copper, and wood. For aluminium, cement, and plastics, the full implementation of ME will increase global secondary production but for different reasons: much higher recycling rates (plastics), higher in-use stocks of aluminium and thus higher scrap flows, and reuse of concrete elements (cement).

For steel and cement, current in-use stocks per capita differ by a factor of ca. 3 across the two world regions (Fig. 4b). Per-capita in-use stocks of steel and cement converge at the global level, for scenarios with and without ME. This is mainly due to the convergence of per-capita residential floorspace between the Global North and the Global South. Dematerialization in the form of contraction of steel and cement stocks, however, and with it material and GHG savings, are only observed for the ME scenarios. Here, per-capita in-use stocks reflect a global state of service equality and converge to a level that lies in between toady's stock levels in Global North and Global South by the end of this century. For in-use stocks of wood, the difference between the two regions is much smaller. As wood benefits from material substitution, there is no contraction of in-use stocks. Aluminium, copper, and plastics in-use stocks show few signs of convergence. Global North aluminium stocks are not much impacted by ME, because the effects of decreasing product stocks are largely compensated for by the increased aluminium intensity through material substitution. In-use stocks of copper and plastics in the Global North decrease by about one-fifth due to ME, mainly because of the smaller vehicle fleets in the car-sharing and ride-sharing scenarios. Global South stocks of aluminium can increase by up to 70% under material substitution scenarios. As material stock size is determined by several factors, including service demand, technology types, product size, material choice, and ME, no universal trend for the evolution of the different stock curves can be observed for these materials. This means that in order to understand future material stock and production trajectories, models with high technological detail are needed. Scenario studies for changing stock patterns of such materials must take into account such detail to produce consistent and technologically feasible results, rather than assuming simple growth or de-growth patterns for material stocks, which has so far been the case.

This study provides a detailed assessment of ME strategies in two major end-use sectors with a global scope and in a changing socioeconomic and energy-supply context. The high-resolution material and product-life cycle model allows us to quantify the overall impact of ME strategy bundles at scale, taking into account both the mutual dependencies among strategies (e.g., product light-weighting means that less material is available for recycling) and the development of service demand over time. To quantify these effects, our model captures the interaction of product design and life cycles, of material-cycle dynamics, and of macro-level changes of service demand and in the energy system. It hence demonstrates how detailed knowledge about technological change can be relevant for, and used in, global assessments. Material-cycle modelling is largely absent from integrated assessment models, which are the work horses of global climate-mitigation assessment, and assessments such as the one presented here can be soft-linked to and possibly integrated into such models similar to how land-use modelling has recently been integrated. Soft-linking would help establish the stock-flow-service nexus[39], ME strategies and material cycle and resource constraints in climate-mitigation scenarios[40], and integration would allow for including ME into optimization routines. Better integration into large-scale assessments would also allow us to study the global economic implications of ambitious ME.

Although the resource-efficiency and climate-change (RECC) framework features substantial service provision and engineering detail, it needs verification and improvement based on high-resolution product and process-level data. For example, building archetype models including specific components (heating system and plumbing) or process models of waste sorting and scrap remelting[41] should inform changes of parameters in the RECC scenarios in the future. The RECC results represent estimates of the technical potential of ME. To estimate the feasible potential of ME under different business models and policy scenarios, material production and recycling costs need to be included, among others. Adding the cost layer to the material cycles would allow for circular-economy business model simulation for ME[42] and the estimation of employment impacts[43]. Combined with macro-economic modelling, cost information would enable us to quantify rebound effects[44] due to lower material prices from under-utilized primary production assets and increased availability of (lower quality) recycled material[45]. Including costs would facilitate the simulation of policies to mitigate ME rebounds, such as eco-design standards, cap and trade systems for recourses, or raw material extraction taxes.

The findings confirm that, for deep emission reductions in the residential-building sector, low-carbon electricity by itself will not be sufficient[46], but additional demand-side efficiency and sufficiency measures are required[47]. The same holds for the vehicle fleet, where electrification and a transformation to low-carbon electricity must go hand in hand, as confirmed by our results. Lifting ME to similar prominence as energy efficiency increases the feasibility of attaining the Paris goal of limiting global warming to well below 2 °C and may reduce the dependency on negative emission technologies. As countries struggle to implement and update their nationally determined contributions to the Paris climate agreement, new mitigation options, and co-benefits with other sustainable development goals are needed to get them on track[48]. ME shows strong co-benefits in savings of raw materials, energy, and GHG emissions, and its technical and scaling feasibility is high. These advantages over negative emission technologies represent a compelling reason to give ME a higher priority in climate policy.

**Table 1 RECC v2.4 data aspects and their resolution.**

| Model and data aspect | Resolution |
|---|---|
| Time | 2016–2060 in steps of 1 year, results are reported for/by 2050. |
| Age-cohorts/ Vintages | Vehicles, 1980–2060; residential buildings, 1900–2060 |
| Regions | 20 Countries and world regions, covering the entire world, with the following aggregation: Global North: OECD countries, countries of the former USSR, and China. Global South: India, Africa, Latin America, Middle East, All other Asian countries. G7: Canada, France, Germany, Italy, Japan, UK, USA. |
| Products | 6 Passenger-vehicle drive technologies (with a total of 48 vehicle archetypes), 13 residential-building types (with a total of 52 archetypes) |
| Engineering materials | Construction grade steel, automotive steel, stainless steel, cast iron, wrought Al, cast Al, copper electric grade, plastics, wood and wood products, zinc, cement, and concrete aggregates |
| Waste and scrap types | Heavy melt, plate, and structural steel scrap; steel shred; Al extrusion scrap, auto rims, clean; Al old sheet and construction waste; Al old cast; copper wire scrap; construction waste, concrete, bricks, tiles, ceramics |
| Chemical elements | C, Al, Cr, Fe, Cu, Zn, 'other', traced through materials and waste |
| Energy carriers | Electricity, coal, hard coal, diesel, gasoline, natural gas, hydrogen, fuel wood |
| Service categories | Driving (vehicles); heating, cooling, domestic hot water (residential buildings) |
| Socioeconomic scenarios | Three socioeconomic scenarios: low energy demand (LED) with substantial rollout of demand-side solutions and sufficiency[25], and two shared socioeconomic pathways[26] SSP1 (easy mitigation of and adaptation to climate change) and SSP2 (moderate mitigation and adaptation) |
| Climate policy scenarios | Each socioeconomic scenario is combined with two climate policy scenarios: no new policy after 2020 (reference scenario 'No Pol.') and a policy mix compatible with 2 °C (66% probability), corresponding to RCP2.6 forcing pathway (energy supply GHG mitigation scenario '2 °C Pol.'). The 2 °C policy mix includes higher heating system efficiency, more building renovation, lower GHG intensity of energy supply, more material production decarbonization and energy efficiency, and electrification of the vehicle fleet and the building stock. These parameter values are region-specific and documented in the parameter database and the supplement. |

A complete list of all model aspects and their resolution is available in the model documentation (Section 3.5 of the Supplementary Material).

## Methods

The analysis shown here is based on the RECC model framework[49], which starts with a given demand for services (individual motorized transport and shelter) and calculates the necessary size and operation of in-use stocks of products (passenger vehicles and residential buildings) to deliver these services. RECC routines for stock-driven modelling[50] are then used to translate product in-use stock demand into production of new and recycling of old vehicles and buildings. RECC thus represents a high-resolution implementation of the stock-flow-service nexus[39] and of central parts of the energy service cascade[51] (Supplementary Fig. 1). Both concepts describe that material consumption serves the purpose of maintaining and expanding service-providing in-use stocks, and that each step of the cascade (Supplementary Fig. 1) represents an opportunity to decouple human well-being from the negative environmental impacts of resource extraction and material production.

A key innovation of RECC is the upscaling of representative single-product descriptions ('archetypes') with different degrees of material and energy efficiency. The product archetypes were simulated with engineering tools that model building energy balance and vehicle driving cycles. Using historic product-material-composition data and the product-archetype descriptions, RECC converts the flows of new and old products into material flows, thus representing a dynamic material-flow analysis[52,53]. It also calculates all energy demand from material production, recycling, and product operation, and estimates related GHG emissions via environmental extensions, similar to life cycle impact assessment[54].

With this setup, RECC generates a set of what-if scenarios[55] for different, exogenously controlled degrees of ME in the vehicle and building sectors, and the related major material cycles against different socioeconomic, technological, and climate-political backgrounds[28]. RECC does not assess the likelihood of realization of any of the scenarios studied but checks if mass balance constraints (e.g., by long product lifetimes or limited scrap supply) render some scenarios unfeasible from a material-cycle point of view.

**RECC v2.4 model resolution.** The different system aspects (time, age-cohort, region, etc.) covered by the version of the RECC model developed for this work are shown in Table 1.

**The RECC database.** The RECC v2.4 database contains 104 model parameters. Parameters range from static values (direct emissions of combustion per MJ of energy carrier) to highly detailed and uncertain, and thus scenario-dependent datasets (e.g., the future energy-carrier split of buildings by region, time, and demand (heating/cooling/hot water)). The RECC database has a comprehensive scope and was compiled as a community effort involving many experts. Data templates and project-wide classifications were used to facilitate the compilation of the various types of information. Depending on data availability, we applied several pathways of data compilation:

- Extract mostly socioeconomic parameters from existing scenario models (scenario reference)
- Compile own plausible scenario estimates for socioeconomic parameters in line with the different scenario narratives where results from established model frameworks are not available (group-consensus scenarios)
- Extract process-, product-, and material-specific data from the engineering and industrial ecology literature ('bottom-up' data)
- Extract quantitative estimates of resource-efficiency-strategy potentials, mostly related to prototypes and case studies, from the literature (strategy potentials)
- Simulate energy consumption and material composition of building and vehicle archetypes with specialized software, which are then used as bottom-up product descriptions with and without implementation of ME strategies (product-archetype descriptions).

Data were parsed and reviewed by the RECC team, then aggregated, disaggregated, and/or interpolated to fit the project-wide classification. For each parameter file, the data-gathering process is documented both in the respective template files in the RECC database (whenever only Excel was used) and in custom scripts (for more comprehensive datasets that required pre-processing).

**Scenario reference and group-consensus storyline extension.** We define future scenarios for passenger-vehicle and residential-building operation by augmenting the storylines of the shared SSPs[26,30] to describe future service demand in the passenger-vehicle fleet and residential buildings, and calculate associated material requirements, covering the entire globe in 20 countries/regions until 2060. Data from the World Energy Outlook and Energy Technology Perspectives models were used for the share of electric and hybrid vehicles or building energy mix[56]. The GHG mitigation potential of ten ME strategies at different stages of the material cycle is quantified by ramping up their implementation rates to the identified technical potentials by 2040. Each ME strategy can be implemented separately or as part of a cascade of strategies. The model allows for calculating the impact of one strategy at a time (which is mostly used for sensitivity analysis) or a bundle of strategies in different orders of implementation, each for different socioeconomic and climate policy scenarios.

For some parameters such as the future stock levels or the split of residential buildings into different types, no detailed SSP-consistent scenario calculation was available to which we could refer. Hence, we assumed a set of plausible target values for a number of socioeconomic parameters in line with the storylines of the individual socioeconomic scenarios, as documented by Fishman et al.[30]. This process has been used when translating broad storylines into parameters with high product and regional resolution, and sector specificity (see refs. [25,28]).

**Bottom-up data on technology.** For the energy intensity, emission intensity, and material composition of products and processes, detailed but representative product or process descriptions were compiled from the literature and available

**Table 2 RECC Global main input data and assumptions.**

| | LED | SSP1 | SSP2 |
|---|---|---|---|
| 2050 World population (million) | 8937 | 8246 | 8937 |
| Global North average passenger-km/yr 2016/2050 | 2016: 6360<br>2050: 5990 | 2016: 6470<br>2050: 8600 | 2016: 6570<br>2050: 10,800 |
| Global North average heated residential building m²/cap 2016/2050 | 2016: 27.3<br>2050 (No ME): 28.3<br>2050 (Full ME): 28.3 | 2016: 27.5<br>2050 (No ME): 40<br>2050 (Full ME): 31.5 | 2016: 27.7<br>2050 (No ME): 48.5<br>2050 (Full ME): 39 |
| Global North average cooled residential building m²/cap 2016/2050 | 2016: 28.8<br>2050 (No ME): 19.4<br>2050 (Full ME): 19.4 | 2016: 28.9<br>2050 (No ME): 30<br>2050 (Full ME): 24.5 | 2016: 29<br>2050 (No ME): 42<br>2050 (Full ME): 33.6 |
| Global South average passenger-km/yr 2016/2050 | 2016: 970<br>2050: 1610 | 2016: 1100<br>2050: 4050 | 2016: 1200<br>2050: 5900 |
| Global South average heated residential building m²/cap 2016/2050 | 2016: 11<br>2050 (No ME): 18<br>2050 (Full ME): 18 | 2016: 11<br>2050 (No ME): 22<br>2050 (Full ME): 18.6 | 2016: 11<br>2050 (No ME): 27.3<br>2050 (Full ME): 22.1 |
| Global South average cooled residential building m²/cap 2016/2050 | 2016: 12.3<br>2050 (No ME): 13.8<br>2050 (Full ME): 13.8 | 2016: 12.4<br>2050 (No ME): 22<br>2050 (Full ME): 18.7 | 2016: 12.4<br>2050 (No ME): 33<br>2050 (Full ME): 26.7 |
| *Material efficiency: (I): Industrial material efficiency, (D): Demand-side material efficiency* | | | |
| Material efficiency buildings, full implementation by 2040 | | | |
| End of life recovery (I) | 95% Recovery of steel and aluminium, 93% copper, 70% plastics | | |
| Fabrication yield loss (I) | Decrease to 10% | | |
| New scrap diversion (I) | Up to 80% of all fabrication scrap is used without re-melting | | |
| Reuse at end of life (I) | +29% Steel reuse, +27% concrete reuse | | |
| Lifetime extension (D) | Lifetime extended by 90% | | |
| Material substitution (I) | 85% Of new buildings | 50% Of new buildings | 10% Of new buildings |
| Less material by design (I) | 85% Of new buildings | 55% Of new buildings | 35% Of new buildings |
| More intense use (D) | None (baseline) | −20% Of m²/cap, ≥LED | −20% of m²/cap, ≥LED |
| Material efficiency vehicles, full implementation by 2040 | | | |
| End of life recovery (I) | 95% Recovery of steel and aluminium, 82% copper, 70% plastics | | |
| Fabrication yield loss (I) | Decrease to 10% | | |
| New scrap diversion (I) | Up to 80% of all fabrication scrap is used without re-melting | | |
| Reuse at end of life (I) | 20–40% Reuse | 20–40% Reuse | 9–20% Reuse |
| Lifetime extension (D) | Lifetime of PHEV, BEV, FCV extended by 20% | | |
| Material substitution (I) | For 60% of new vehicles | For 60% of new vehicles | 28–35% Of new vehicles |
| Downsizing (D) | Share of microcars and passenger cars 80–96% | Share of microcars and passenger cars 70–95% | Share of microcars and passenger cars 65–94% |
| Car-sharing (D) | 30% Service demand through car sharing | 25% Service demand through car sharing | 15% Service demand through car sharing |
| Ride-sharing (D) | 40% Increase in occupancy rate | | |
| Climate policy parameters | | | |
| GHG intensity of electricity generation, global average | 241 g $CO_2$-eq/kWh (no new policy), 87 g $CO_2$-eq/kWh (2 °C climate policy) | | |
| Global average share of electricity and $H_2$, vehicles, 2016/2050 | 2016: 0%, 2050: 6–7% (no new policy), 39–40% (2 °C climate policy) | | |
| Global average share of electricity and $H_2$, residential buildings, 2016/2050 | 2016: 33%, 2050: 64–66% (no new policy), 67–70% (2 °C climate policy) | | |

databases. These data include the material composition and specific energy consumption of vehicles and buildings[57–59], e.g., the loss and recovery rates for the manufacturing and waste-management industries[54,60], and the specific energy consumption and process emissions for the manufacturing, waste management, and primary material production industries[56,61]. Although the data can be regarded as representative of current average global technology, their main limitation is that they are static and no information on their change under different socioeconomic and climate policy scenarios is given. To become more realistic, a scenario reference was made wherever possible (cf. above), e.g., for the changing GHG intensity of the supply of different energy carriers, for which a combination of MESSAGE IAM[28] results and IEA Energy Technology Perspective[56] results was used. For the average GHG intensity of primary metal production, emissions from ecoinvent[61] were updated to take into account scenario-dependent changes of the GHG intensity of electricity generation.

For some ME parameters, including the improvement potentials for fabrication scrap, end-of-life recovery efficiency of scrap, reuse of steel and cement components in buildings, or product-lifetime extension, previous estimates can be used[54,62,63].

**Building- and vehicle-archetype descriptions**. Here, 'archetype' refers to an idealized representative and scalable description of the physical properties (energy

intensity of operation and material composition) of a product with a certain functionality, assuming typical user behaviour in a given region. For passenger vehicles, drive technology, segment (car size), and material design choice together determine the archetypes' material composition, and the three properties above plus the assumed driving cycle determine its specific operational energy consumption (specific = per km driven).

For residential buildings, the building type, energy standard, material intensity (conventional or light-weight design), material design choice, and stylized climate conditions (heating and cooling degree days by region) together determine the archetypes' material composition and specific operational energy consumption (specific = per m² and year).

For the final product categories, residential buildings and vehicles, the product-specific simulation tools EnergyPlus (https://energyplus.net/), BuildME (https://github.com/nheeren/BuildME), GREET (https://greet.es.anl.gov/), and FASTSim (https://www.nrel.gov/transportation/fastsim.html) were used to model the archetype descriptions by deriving model estimates for both the material composition and energy intensity of operation for different building and vehicle configurations. For each of the nine building and six vehicle drive technologies, four different archetypes, representing maximal potential for change, were simulated as follows: a standard product without special consideration of ME, a downsized product, a product with ambitious material substitution, and a downsized material-substituted product.

**Overview table with main assumptions**. Table 2 shows the main quantitative assumptions behind the RECC parameter datasets. For a complete overview, we refer to the RECC scenario publication[30] and database (https://doi.org/10.5281/zenodo.4671643).

The RECC system variables and model equations are listed and explained in Section 6 of the Supplementary Material.

## Data availability

The input data and model results of this study were deposited on Zenodo with hyperlinks https://doi.org/10.5281/zenodo.4671643 for the input data and https://doi.org/10.5281/zenodo.4698619 for the output data. Both datasets were released under a permissive license. Source data for the figures are provided along with this paper as supplementary data file. The supplement contains a number of diagrams for central input data and results. Source data are provided with this paper.

## Code availability

The complete model code is open source and modular. Third parties can modify the scenario assumptions and run calculations with custom parameters and scenario storylines. A detailed description and definition of all model aspects, the classifications used for them, the system variables and parameters, the model equations and their division into modules and the data compilation, (dis)aggregation, and formatting process are contained in the model documentation in the Supplementary Material. The model code is available on GitHub via https://github.com/YaleCIE/RECC-ODYM (commit f2f59c4) and https://github.com/IndEcol/ODYM (commit 3f65f98).

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

## Acknowledgements
S.P. was supported by grant number 7635.521(15) from the Ministry of Science, Research and the Arts of Baden Württemberg (Germany). T.F.'s work was supported by the Israel Science Foundation (grant number 2706/19). We received support from the United Nations Environment Programme for database compilation and model development. We thank Farnaz N. Asghari, Lucca Ciacci, Beijia Huang, Aishwarya Iyer, Eric Masanet, Koichi Kanaoke, Stefanie Klose, Volker Krey, Zeina Najjar, Thibaud Pereira, Laurent Vandepaer, Paula Vollmer, and Dominik Wiedenhofer for contributing to the project's open database.

## Author contributions
S.P., N.H., T.F., and E.H. designed the research. N.H., S.P., and E.H. managed the project workflow. T.F. designed the scenario formulation approach and—together with N.H. and the rest of the team—ensured overall scenario consistency. Q.T. and P.W. contributed to the modelling of the vehicle archetypes and scenarios. N.H. developed the building archetype model. N.H., A.N., and P.B. contributed building archetype and scenario formulations. S.P. implemented large parts of the model framework, compiled the database, and conducted the scenario calculations. All authors contributed to analysing the results and writing the paper.

## Funding

## Competing interests
The authors declare no competing interests.
