## [Peer Review File · Nature Communications]

REVIEWER COMMENTS

Reviewer #1 (Remarks to the Author):

Global Scenarios of Resource and Emissions Savings from Systemic Material Efficiency in Buildings and Cars

Stefan Pauliuk, Niko Heeren, Peter Berrill, Tomer Fishman, Andrea Nistad, Qingshi Tu, Paul Wolfram, and Edgar G Hertwich.

This paper presents an assessment of the potential GHG emissions reductions from applying material efficiency saving in residential buildings and cars. The paper is well written and includes an extensive Supporting Information document.

> General comment

My primary concern, after reading the paper, is the lack of distinction about the novelty of the paper, in relation to other papers and reports published by the same authors. I list below four examples, that I know of, of recent literature, which seems to cover similar ground, using the same models and data sets. This is not to say that this paper has not added significant new knowledge above these papers, but it is not clear in the paper which parts of the assessment are new (data, model, approach) and which are simply cranking the handle on existing models. This is a real concern for the novelty and authenticity of this paper.

Can you please be explicit about what this paper adds over and above the other papers/reports? And clearly delineate the modelling work and data used for this paper, from the previous modelling work already described in other papers and reports.

- Hertwich, E. G. et al. Material efficiency strategies to reducing greenhouse gas emissions associated with buildings, vehicles, and electronics—a review. *Environmental Research Letters* 14, 043004 (2019).
- Fishman, T et al. A comprehensive set of global scenarios of housing, mobility, and material efficiency for material cycles and energy systems modelling (submitted to JIE).
- Hertwich, E.G., R. Lifset, S. Ali, S. Pauliuk, N. Heeren, and Q. Tu. 2019a. Resource Efficiency and Climate Change: Material Efficiency Strategies for a Low-Carbon Future. Summary for Policy Makers. A report of the International Resource Panel. Nairobi, Kenya.
- Hertwich, E.G., R. Lifset, S. Pauliuk, N. Heeren, S. Ali, P. Berrill, T. Fishman, Q. Tu, and P. Wolfram. 2019b. Bridging the gap: enhancing material efficiency in residential buildings and cars. In *Emissions Gap Report 2019*, 56–62. Nairobi: United Nations Environment Programme.

Furthermore, it is not clear what novelty this paper provides over other assessments, such as the Grubler et al 2018 LED scenario paper, which also assessed ME strategies in an integrated way. Perhaps your assessment of house and cars is more accurate, using a bottom up model ... if so then focus on describing this advancement in knowledge. And compare it back to other similar ME assessments. I am struggling to see what additional insight this paper is providing.

>Title

- The title states “buildings” but I see no evidence that commercial or industrial buildings have been analysed in the paper. Please correct to residential buildings or houses.
- I question the use of ‘systemic’ in the title, given this term is not used or explained in the remaining paper. In what way is the assessment systemic? (I only covers houses and cars, which is a small part of the entire materials system).

>Abstract

The opening statement of the abstract "Material production now accounts for 23% of global greenhouse gas (GHG) emissions" seems disconnected with the paper, and is also controversial, without much qualification. It is stated as a fact, and yet the figure of 23% derives from a recent analysis by one of the authors (Hertwich, 2020, preprint), where the percentage is specific to a particular definition of "materials", and the allocation of carbon emissions along supply chains uses the allocation of carbon emissions using financial transactions in a MRIO methodology. The use of IO for allocating emissions is subjective by nature, because financial flows do not always track physical reality, and is widely disputed in academic communities. So leading with this statement, which can only be qualified by reading the other paper, immediately distracts from the paper. The statement adds little to the paper but acts like a red rag to a bull for other academics working on industrial emissions.

> Main text

L128 Please explain what the "expert consensus approach" means, and what this methodology consisted of. I find no reference to this in the Methods or the SI. It sounds like a few friends got together to estimate the future service levels.

Fig 2:

- It is not clear how much of each of the ME strategies has been applied, and what assumptions are implicit in the values shown. This needs to be in the main paper, not the SI. It is hard to judge whether these are credible numbers, without seeing how much reuse, higher yields, etc, has been implemented. If the assumptions are from other papers, then I don't see why this Figure is included.
- the coding for the scenarios (i.e. NoPoI, RCP2.6) has not been explained.
- Why are the bars all the same height, when the amount of cumulative and annual emissions in each scenario is very different. This mean you can't compare the ME saving strategies (by colour) across the scenarios.

Fig 3:

- It is not clear, which of the solid curves (or dotted curves) is primary or secondary.
- Assuming the darker lines are primary, then it seem that secondary production of steel, aluminum, and copper overtakes primary production. Have the dynamics of this been taken into account. The only way for secondary production to overtake primary is for the overall demand to be reducing, and for all lag in the system due to lifetimes to have caught up. It doesn't seem from a simple view, that these conditions have been achieved. Please explain.

Fig 4

- Why are "energy efficiency and low carbon energy supply are introduced first, and ME strategies are then applied on an already decarbonized system ...". There is no obvious logic for applying EE and decarb preferentially to ME, or vice versa. Why not model the application of both at the same time, averaging the saving between them. This could be achieved using standard sensitivity analysis approaches.

L245: "This study provides the first comprehensive global assessment of ME strategies in a changing socioeconomic and energy supply context." I'm not sure you can be this bold. Firstly, your ME strategies are for passenger vehicles and residential buildings, which means it is not comprehensive. In addition, the Grubler et al. (2018) paper also did a global assessment of ME strategies (all, not just cars and houses) under changing socioeconomic and energy supply contexts.

L254: "While service, product, and material cycle modeling is largely absent from integrated assessment models," Again, some mention of the Grubler et al. (2018) paper is surely required here,

as they have integrated the ME assessment into the IAM modelling (albeit exogenously).

>Materials and Methods

This section describes several methods and databases, including RECC, stock-flow-service-nexus, SSP, WEO, ETP, MESSAGE IAM, etc. Alongside this, it refers to data and analyses from numerous other papers. Furthermore, Table 1 gives only a generic description of the methodological approach. I am afraid this reads more like a literature review of models, than a method section of what you did for this paper. And it is unclear whether what actually modelling work was completed for this specific paper (as opposed to the many others listed by the authors). I suggest this needs a complete rethink to clearly delineate between the models and data you use which has already been published, and the additional research completed for this paper which is not already published. Without this, it is difficult to ascertain the novelty of this paper.

Reviewer #2 (Remarks to the Author):

In this manuscript, the authors present a comprehensive dynamic Material Flow Analysis, with GHG emissions link, for passenger vehicles and residential buildings from 2016 to 2060. They do this to compare the effect of GHG emissions reduction potential of Material Efficiency strategies with climate policy strategies. Analyzing co-benefits between circular economy and climate neutrality targets is highly relevant research. The extensive model ODYM-RECC, utilizing the newly developed ODYM framework, quantifying results for 20 world regions and considering material and energy flows in 16 processes including a use-phase of various car types and building types ("archetypes") is definitely new in peer-reviewed research. Parts of the results and model have been published in the International Resource Panel report on material efficiency and similar publications, or as partial preprint documents. The results are interesting, because they highlight the existence of anticipated co-benefits, the need for material efficiency measures to reach anything close to climate neutrality and the robustness of these results based on different scenarios including Shared Socioeconomic Pathways. In particular, the transparency and accessibility of this research project is remarkable.

Major comments:

A) Time frame: I'm missing a good justification for choosing 2060 as last year in the dMFA and 2050, in contrast, as year for evaluation, e.g. in Fig. 2. It is a bit confusing that the abstract talks about GHG emissions "until 2060", but the results section shows annual and cumulative results for 2050. A reason for different reference year, or extending the model calculations beyond the main year of interest (2050) is not given. You should consider either harmonizing, or better justifying the choice of time frames in your article.

B) Archetype description and results display: I see a potential mismatch between vehicles and buildings in your model, at least in the way you show the intermediate results for vehicle and building stock in the Supporting Information. While the vehicles are shown disaggregated for fuel type, the buildings are shown disaggregated by energy efficiency standard and number of households in the building. If I understood your model correctly, then these are not the "archetypes" of vehicles and buildings in your model, but they are energy efficiency measures. I think it would be more intuitive if the results for the building stock were shown in terms of heating fuel type and cooling device, rather than building size, because this would be the equivalent to vehicle fuel type. And heat pumps, for example, are only mentioned very briefly in l. 159 in the main manuscript. The equivalent to building size would be car type (micro car to light truck), which however is part of your archetypes. Therefore, I see a need to clarify which parameters go into archetypes, which parameters are energy efficiency measures and which parameters are material efficiency measures.

C) Current trajectory of policies: You model three different pathways for society (LED/SSP1/SSP2). Your model starts in 2016, but we are already in late 2020. Could you mention, anywhere, which path we are currently following? I'm most familiar with GHG emissions from passenger vehicles and from what I see in Fig. S-7, I would say we are much closer to SSP1 and very far away from LED scenario. If you have data on this, this could be worth a critical note in your manuscript.

D) Public transportation: Looking at Fig. SI4-1 (Drivers and in-use stock parameters, USA), I'm curious whether the assumed difference in passenger kilometres (which is essentially the functional unit) comes from a shift in modal split, or from a reduced mobility demand. If the first one is the case, then there should be at least a discussion whether infrastructure expansion for public transportation might lead to trade-offs in terms of material efficiency and GHG requirements. At this moment, public transportation seems to be completely outside of your model and this not discussed anywhere in the manuscript. In the end, the most effective ride-sharing option is public transportation, isn't it?

Minor comments to the manuscript:

- 1) L. 56ff: In the last sentences of the abstract, the order of passenger vehicles and residential building is reversed, should be the same.
- 2) L. 60: Last sentence should end with "for passenger cars" instead of "for residential buildings".
- 3) L. 67: What does "rate-limited" mean in this context? Expression is not clear to me.
- 4) L. 70: Typo: "of" instead of "if".
- 5) L. 73: Why is this a competition? Can't we do both decarbonizing material production and low-carbon energy provision?
- 6) L. 115: Would be helpful to name SSP1 and SSP2 here. In Fig. 4.1 of the Documentation, this is "Low Challenges and Intermediate Challenges".
- 7) L. 154: You introduced the terminology of Global North and Global South, so it's better to stick to that other than using the term "developing countries".
- 8) L. 170: Isn't the additional emissions savings of 1-2 Gt/yr too big? Total reductions from resource efficiency are at max 2 Gt/yr in Fig. 1.
- 9) L. 182-189: The whole paragraph uses unnecessarily complicated sentences and the meaning overall is not clear. Try to simplify your expressions. If I understood your model correctly, the baseline for 2050 annual GHG emissions varies between 1.8 Gt (LED+RCP2.6) and 15 Gt (SSP2+NoPol). From this baseline, the ME-induced reductions for 2050 annual GHG emissions cut between 1.1 Gt (SSP2+NoPol) and 3.4 Gt (LED+RCP2.6). This means the lower the baseline, the higher the relative additional emission reductions from ME (61% for LED+RCP2.6 and 22% for LED+RCP2.6). And the relative GHG reductions are bigger for 2050 annual emissions than for cumulative emissions until 2050, because currently the baseline is still high. Is that correct?
- 10) Fig. 2: This figure should be improved: 11800 Mt can not be read well. It is not clear enough that the first number applies only to the brown Bottom line. Capitalization in the legend is inconsistent. The ME measures listed in the legend are not described, or even mentioned in the text so far.
- 11) L. 334: Typo: "product-" instead of "product".
- 12) L. 352: I guess the comma should read "or" instead. It is either Excel sheets or scripts. Or, alternatively, did you miss writing about a third option?
- 13) Table 1: Does "other" refer to wood and plastics? Why are they not modelled as wood and plastic separately?
- 14) L. 411: Either "and " or "to" are not needed here.
- 15) L. 555f: "A.N." appears twice in the list for archetype and scenario formulations.
- 16) L. 479: Ref. Pauliuk et al. 2020 volume, issue, pages need update.

Minor comments to the Supplementary Material 1:

- 17) Figure numbers: There is a mismatch between numbers given until Fig. SI4 (no hyphen) and starting with SI-5 (with hyphen). In section 4.6 it's all of a sudden SI1-15 (and so on) instead of SI-15. Please align the numbering and formatting.

- 18) Figs. SI-5-1 to SI-6-20: Changes between top row and bottom row would be much easier to grasp if y-axis limits were chosen the same. Also it would be good if title or axis labels, or legend, mentioned the difference in "no ME" and "full ME" spectrum. The reader should not have to search for this important information in the figure caption.
- 19) Fig. SI-5-4: Why is there a significant share of HEV in all European regions, but not in Germany? Seems inconsistent.
- 20) Figs. SI-5-1 to SI-6-20: I don't get the order of countries. It's alphabetical, not by size of the country and not by world region either. With 20 regions, this should follow some intuitive logic and should be consistent throughout the manuscript and supplementary.
- 21) Figs. SI-7, SI-8: Please split this into three figures each, so that caption and figure are always on the same page.
- 22) Figs. SI-7, SI-8: Please don't just write pav and reb in the figure titles, write "Passenger vehicles" and "Residential buildings" instead.
- 23) Figs. SI-7, SI-8: The order of countries in the 5x5 subplots changes between the figures. As mentioned above, the order of countries should always be consistent and follow some logic.
- 24) Fig. SI-9: Why does such a figure exist only for passenger vehicle sector? Is the buildings figure missing?
- 25) Fig. SI-9: Please don't use bar graphs with logarithmic scale axis. In particular not stacked bar graphs. It can lead to the case that smaller values get more inked area, which is counter-intuitive and therefore should be avoided in any graphical display of data.
- 26) Section 4.3: Don't use pav and reb abbreviations in section headings.
- 27) Fig. SI-11: Again, the Figure caption should not be the only place to distinguish left and right plots. They have exactly the same titles, labels and legends. Please add any information on baseline and RCP2.6 to these graphs.
- 28) Fig. SI-11: I'm confused by the passenger vehicle graphs because of (most likely) negative values in a stacked bar graph. The dark colour does not appear in the legend and the order of elements seems to change.
- 29) Section 4.6: Is this primary energy or final energy consumption?
- 30) Fig. SI1-23: The exponents (1e8 and 1e7) are somewhat distracting and overlapping with the other graphs.
- 31) Fig. SI1-24: Units are not clear. Passenger-kilometer per GHG kilometer/tonne? What is a GHG kilometer?
- 32) Ref. Reyna et al. 2014: Volume, Issue, Pages wrong
- 33) Ref. Riahi et al. 2017: Volume, Issue, Pages missing

Minor comments to the ODYM-RECC Documentation:

- 34) Box 1: "System boundary" should not be capitalized.
- 35) Table 1.1: CaS and RiS should only apply to vehicles and MIU should only apply to buildings, if I'm not mistaken.
- 36) Fig. 3.4: Why are % split in pass. Vehicles, trains, bus, etc. and % split in to single and multi-family houses, apartment blocks "currently not implemented"? What does this mean to the model? Don't you show exactly such a building type split in the Supplementary Material 1?
- 37) Table 3.1: Maybe just a comment for the future: What are you going to do if your model, at any point in time, exceeds 52 indices? Is there any possibility to extend index letters beyond this point? (Probably needs to be addressed in ODYM, not ODYM-RECC).
- 38) P. 26: See above, what is "other" elements?
- 39) Fig. 4.2: Because you are using SSP1 and SSP2 in the model, why do you show SSP2 and SSP5 here?
- 40) Ref. Bürger et al. 2018: Published where and how?
- 41) Ref. Deetmann et al. 2019: Volume and Issue missing
- 42) Ref. Deetmann et al. 2018: Issue and Pages missing

- 43) Ref. Elshkaki et al. 2018: Volume, Issue, Pages missing
- 44) Ref. Haberl. Et al. 2017: Pages missing
- 45) Ref. Hertwich et al. 2015a/b looks like the same publication.
- 46) Ref. Laner et al. 2014: Volume, Issue, Pages wrong
- 47) Ref. Nakamura et al. 2017: Please refer to the ES&T publication, not a preliminary draft.
- 48) Ref. Riahi et al. 2017a/b looks like the same article
- 49) Ref. van der Voet et al. 2018: Volume, Issue, Pages missing
- 50) Ref. Winning et al. 2017: Volume, issue, Pages missing

Overall, I expect that the authors will be able to address these issues. I recommend inviting the authors to revise their manuscript.

Christoph Helbig

Reviewer #3 (Remarks to the Author):

Authors present a comprehensive global assessment of Material Efficiency strategies for climate mitigation. This is a timely contribution, given the high policy priority around resource efficiency and climate change. One of the unique aspect of this study remains the methodological combination of design parameters and product life cycles with operational and embodied climate impacts for residential buildings and passenger vehicles. From practice point of view, all possibilities of major material efficiency strategies have been included in the assessments. There is an adequate justification for specific materials and scenarios.

From a methodological perspective, I have a few comments for authors:

One of the major challenge in this research, as I see, is associated with the huge range of expected benefits and a significant uncertainty in the estimates. For example, given the global nature of coverage- one may argue that it may be possible to create limited archetypes for passenger vehicles but the material intensity factors and operational energy for residential buildings would take more then the archetypes one can ideally agree upon. How do authors see these issues influencing the results?

I can see that a very limited number of climate mitigation assessments of material efficiency strategies have informed the global estimates of potential emission savings in this study. Do authors find these sources sufficient to generalize impact values for all countries?

Even though technology and energy aspects over product life cycles have been combined in this study, Behavioral aspects of firms and users have been missed out of the equation. User behavior is arguably one the biggest influence in the material efficiency adoption at scale. Choice of sharing a vehicle or reusing a window or building component is highly consumer dependent. Would it be wise to have scenarios influenced by the consumer acceptance?

Some minor comments:

Line 60- passenger vehicles instead of residential buildings?

Line 70- 'now 23% if' Is this statement still true accounting for the expected decline in GHG emissions in 2020? It would be helpful to mention the year or remove 'now'.

Line 78- Supply-side mitigation seems ambiguous term and the cost reduction may not necessarily be true given the complexity of trade-offs.

Line 80- 'Reducing material use' seems ambiguous. Would it be more appropriate to say that these measures are to reduce material demand and increase/maximize the material use?

Line 86- re-use of scrap seems to included in both fabrication efficiency and waste management- can

authors clarify this aspect?

Line 177-180 In assessing the annual and cumulative GHG emissions savings, what is the expected scale of adopted ME measures? Full ME adoption- does it include all ME strategies adopted by complete existing and upcoming stocks of buildings and vehicles?

Reviewer 1

REVIEWER COMMENTS

Reviewer #1 (Remarks to the Author):

Global Scenarios of Resource and Emissions Savings from Systemic Material Efficiency in Buildings and Cars
Stefan Pauliuk, Niko Heeren, Peter Berrill, Tomer Fishman, Andrea Nistad³ Qingshi Tu⁵ Paul Wolfram, and Edgar G Hertwich.

This paper presents an assessment of the potential GHG emissions reductions from applying material efficiency saving in residential buildings and cars. The paper is well written and includes an extensive Supporting Information document.

> General comment

My primary concern, after reading the paper, is the lack of distinction about the novelty of the paper, in relation to other papers and reports published by the same authors. I list below four examples, that I know of, of recent literature, which seems to cover similar ground, using the same models and data sets. This is not to say that this paper has not added significant new knowledge above these papers, but it is not clear in the paper which parts of the assessment are new (data, model, approach) and which are simply cranking the handle on existing models. This is a real concern for the novelty and authenticity of this paper.

Can you please be explicit about what this paper adds over and above the other papers/reports? And clearly delineate the modelling work and data used for this paper, from the previous modelling work already described in other papers and reports.

1 Hertwich, E. G. et al. Material efficiency strategies to reducing greenhouse gas emissions associated with buildings, vehicles, and electronics—a review. *Environmental Research Letters* 14, 043004 (2019).

2 Fishman, T et al. A comprehensive set of global scenarios of housing, mobility, and material efficiency for material cycles and energy systems modelling (submitted to JIE).

3 Hertwich, E.G., R. Lifset, S. Ali, S. Pauliuk, N. Heeren, and Q. Tu. 2019a. Resource Efficiency and Climate Change: Material Efficiency Strategies for a Low-Carbon Future. Summary for Policy Makers. A report of the International Resource Panel. Nairobi, Kenya.

4 Hertwich, E.G., R. Lifset, S. Pauliuk, N. Heeren, S. Ali, P. Berrill, T. Fishman, Q. Tu, and P. Wolfram. 2019b. Bridging the gap: enhancing material efficiency in residential buildings and cars. In *Emissions Gap Report 2019*, 56–62. Nairobi: United Nations Environment Programme.

Furthermore, it is not clear what novelty this paper provides over other assessments, such as the Grubler et al 2018 LED scenario paper, which also assessed ME strategies in an integrated way. Perhaps your assessment of house and cars is more accurate, using a bottom up model ... if so then focus on describing this advancement in knowledge. And compare it back to other similar ME assessments. I am struggling to see what additional insight this paper is providing.

>> Thank you for giving us a chance to clarify this matter.

Please note that the first paper listed is a review and does not contain any original modeling. The second paper is a description of the methods used in the present assessment. The fourth short report section is a synopsis of results presented in the IRP report. While the present paper is based on the same model as the IRP report, it goes substantially beyond the report in terms of geographical scope (first study with high level of detail AND global coverage) and focus on material stocks and production volumes. In addition, it benefits from another year of model and data development. Please note that it is common practice to use the same model for various research questions of interest. This is the first academic paper which describes the full capabilities of this new model.

We have added following text to the description of the contribution of our paper in the introduction:

L139: “Our analysis covers the resource and GHG impact of ME in these two sectors, covering the entire world comprised of 20 countries/regions, grouped into the Global North (OECD, former USSR countries, China) and the Global South (low and medium-income countries in Asia, Africa, and the Americas). It represents a significant advancement over previous work¹¹ as it covers the entire world and all emerging economies, in particular, and because the scope of the research questions was expanded to quantify the impacts of ME on material stocks and production as well as on the vehicle and building stocks and changes in stocks.“

We also added text to explain the difference to the modeling in the LED scenario:

L108: “The multi-sector low energy demand (LED) scenario²⁵ contains a detailed depiction of end-use energy-related demand-side mitigation strategies. It also includes the mitigation potential of ME but in a very aggregate fashion, by applying a demand-side ‘dematerialization multiplier’ and a supply-side ‘material efficiency’ term. These simplifications limit the ability of these models to accurately quantify the effect of ME on material cycles and related energy use/GHG and thus to identify the most promising strategies. A detailed review of the grey literature assessing ME is contained in section 1 of supplement 1.“

Finally, section 1 of the supplementary material 1 contains a detailed (13 pages) description of the advancement over previous work, the relation to own preparatory work and a detailed comparison to other studies on the system-wide potential and implications of material efficiency.

>Title

- The title states “buildings” but I see no evidence that commercial or industrial buildings have been analysed in the paper. Please correct to residential buildings or houses.
- I question the use of ‘systemic’ in the title, given this term is not used or explained in the remaining paper. In what way is the assessment systemic? (I only covers houses and cars, which is a small part of the entire materials system).

>> The revised title reads “*Global Scenarios of Resource and Emission Savings from Material Efficiency in Residential Buildings and Cars*”, which addresses both concerns. With ‘systemic’, we mean that material efficiency strategies are not only applied to the vehicles/buildings themselves but also to the supplying industries upstream (manufacturing) and downstream (re-use, waste mgt.). But that is probably a bit too much to put it into a single word, so we leave it out in the title and rather explain in the abstract.

>Abstract

The opening statement of the abstract “Material production now accounts for 23% of global greenhouse gas (GHG) emissions” seems disconnected with the paper, and is also controversial, without much qualification. It is stated as a fact, and yet the figure of 23% derives from a recent analysis by one of the authors (Hertwich, 2020, preprint), where the percentage is specific to a particular definition of “materials”, and the allocation of carbon emissions along supply chains uses the allocation of carbon emissions using financial transactions in a MRIO methodology. The use of IO for allocating emissions is subjective by nature, because financial flows do not always track physical reality, and is widely disputed in academic communities. So leading with this statement, which can only be qualified by reading the other paper, immediately distracts from the paper. The statement adds little to the paper but acts like a red rag to a bull for other academics working on industrial emissions.

>> We have slightly modified the wording. Please note that the paper by Hertwich has now been published in *Nature Geoscience*, without receiving any criticism for using MRIO of the sort that the reviewer indicates, neither in the extensive review process or afterwards. In the paper, it is indicated that the results broadly agree with a published bottom-up estimate for a single year based on LCA studies, so that we see these results as fairly robust.

The revised abstract now reads:

“Material production accounts for a quarter of global greenhouse gas (GHG) emissions. Resource efficiency and circular-economy strategies, both industry and demand-focused, promise emission reductions through reducing material use, but detailed assessments of their GHG reduction potential are lacking. We present a global-scale analysis of material efficiency for passenger vehicles and residential buildings. We estimate future changes in material flows and energy use due to increased yields, light design, material substitution, extended service life, and increased service efficiency, reuse, and recycling. Together, these strategies can reduce cumulative global GHG emissions until 2050 by 13–26 Gt CO₂e (passenger vehicles) and 20–52 Gt CO₂e (residential buildings), depending on policy assumptions. Next to energy efficiency and low-carbon energy supply, material efficiency is

the third pillar of deep decarbonization for these sectors. For residential buildings, wood construction and reduced floorspace show the highest potential. For passenger vehicles, it is ride sharing and car sharing.”

> Main text

L128 Please explain what the “expert consensus approach” means, and what this methodology consisted of. I find no reference to this in the Methods or the SI. It sounds like a few friends got together to estimate the future service levels.

>> We revised the text to clarify that expert consensus is only one of several approaches we employ, and add references that provide further detail of our scenario formulation methods:

L164: “On the basis of the LED and SSP scenario storylines,³² we developed parameter values using a combination of data-driven extensions of historical data, literature studies, and expert consensus approaches, similar to the development of the SSP scenarios framework itself. These parameters include future service level (passenger-km delivered by cars, residential floor area utilized) and the share of the different drive and building technologies used. Future service levels were subject to several rounds of consensus building and refinement, documented in detail in an accompanying study.²⁹”

L487: “For some parameters such as the future stock levels or the split of residential buildings into different types, no detailed SSP-consistent scenario calculation was available to which we could refer. Hence, we assumed a set of plausible target values for a number of socioeconomic parameters in line with the storylines of the individual socioeconomic scenarios, fully documented by Fishman et al.²⁹ This process has been used when translating broad storylines into parameters with high product and regional resolution and sector specificity, see Riahi et al.³² and Grübler et al.²⁵”

Fig 2:

- It is not clear how much of each of the ME strategies has been applied, and what assumptions are implicit in the values shown. This needs to be in the main paper, not the SI. It is hard to judge whether these are credible numbers, without seeing how much reuse, higher yields, etc, has been implemented. If the assumptions are from other papers, then I don't see why this Figure is included.
- the coding for the scenarios (i.e. NoPol, RCP2.6) has not been explained.
- Why are the bars all the same height, when the amount of cumulative and annual emissions in each scenario is very different. This mean you can't compare the ME saving strategies (by colour) across the scenarios.

>> We agree and made the following changes:

(1) Fig. 2 was redrawn (see manuscript) so that the scenario abbreviations are fully consistent with what is presented in the introduction, where the scenarios are briefly introduced (more detail in the methods section, full detail in the appendix and accompanying papers).

(2) The reductions here are shown on a % scale, which directly enables the reader to visually compare the relative changes across scenarios, which is the research question addressed here. Absolute changes can be seen from the numbers plotted in the Figure, the at-scale-plots in Fig. 1 (now Fig. 2), and related figures for other scenarios in the supplement.

(3) To be more explicit about the main assumptions, we added a new Table 2 to the methods section (please see the manuscript), where we report the main drivers for the LED, SSP1, and SSP2 scenarios for the Global North and Global South, respectively, and the main potentials (parameter changes) for the ten material efficiency strategies in residential buildings and vehicles in the three scenarios.

(4) We swapped the order Fig. 1 <-> Fig. 2 to create a more plausible storyline: First, total results with strategy detail, then regional patterns.

Fig 3:

- It is not clear, which of the solid curves (or dotted curves) is primary or secondary.
- Assuming the darker lines are primary, then it seem that secondary production of steel, aluminum, and copper overtakes primary production. Have the dynamics of this been taken into account. The only way for secondary production to overtake primary is for the overall demand to be reducing, and for all lag in the system due to lifetimes to have caught up. It doesn't seem from a simple view, that these conditions have been achieved. Please explain.

>> (1) We revised the figure (now Fig. 4) and legend and use a more pronounced color pattern.

(2) Indeed, the gap between primary and secondary production is shrinking and secondary material may even overtake primary production in some sector/region combinations, and we describe and explain this effect in section 5 of the supplement as follows:

“Secondary material production is determined from scrap supply (all available scrap is recycled) and may be exported to other sectors if excessive. Primary production is determined to satisfy demand for stock expansion and for high quality material.”

“Around 2020, material demand for the two sectors is mainly fuelled by primary material production from natural resources. This is because of the still expanding in-use stock, the lack of concrete re-use, and the demand for high-quality primary material especially for vehicles. Over time, the situation changes as stocks grow further, start saturating eventually, and more end-of-life material becomes available. Globally, for the two sectors combined, secondary material availability (determined largely from historic consumption) can overtake primary material demand, meaning that the average recycled content of new buildings and vehicles will exceed 50%. Depending on the degree of ME, this is the case for steel (2025-2048), aluminium (ca. 2025), copper (before 2020), and plastics (ca. 2035 for full ME). Still, some amounts, albeit low, of primary material production are needed even in high ME futures to allow for stock expansion and compensation of dissipative and irreversible losses.”

Fig 4

- Why are “energy efficiency and low carbon energy supply are introduced first, and ME strategies are then applied on an already decarbonized system ...”. There is no obvious logic for applying EE and decarb preferentially to ME, or vice versa. Why not model the application of both at the same time, averaging the saving between them. This could be achieved using standard sensitivity analysis approaches.

>> Thank you for raising the issue. The rationale for the order was that current policies focus on energy and neglect materials, so that we felt it was incumbent on us to show that ME had additional contributions to make that go beyond what is attainable with a focus on energy.

We agree with you that it is instructive to also show the reversed order, showing the impact of ME from the current baseline, and changed the figure to show both sequences (See new Fig. 3 (order was changed)), which provide additional insights:

L265: “The model-estimated contribution of mitigation strategies to overall emission reduction depends on their sequencing. In the bars on the right side of each scenario in Fig 3., energy efficiency and low-carbon energy supply are introduced first, and ME strategies are then applied on an already decarbonized system, yielding higher savings from decarbonization and lower savings from ME than if the sequence was reversed (left side bars). These two alternative sequences show that the impact of ME is larger for SSP1 and SSP2 in a world with high-carbon energy supply, which is a direct consequence of the carbon intensity of material production and of the use phase energy savings mediated by ME. The situation is different for LED, where the GHG savings potential of ME after implementing energy-efficiency and low carbon energy supply is larger than for the opposite sequence, especially for the Global South. The main reason for that effect is that material substitution, which dominates ME GHG savings in LED (see also Fig. 2) becomes much more effective once aluminium production is decarbonized (vehicle steel substitute), which is the case in the right bar but not in the left bar. After seizing the energy-efficiency (green) and energy-supply-transformation (blue) potentials, the share of remaining global emissions reduced through ME is smaller in SSP2 (32%) and SSP1 (39%) than in LED (62%), because ME strategies are applied more gradually and to less ambitious end targets in SSP2 and SSP1, reflecting the storylines of those scenarios.”

L245: “This study provides the first comprehensive global assessment of ME strategies in a changing socioeconomic and energy supply context.” I'm not sure you can be this bold. Firstly, your ME strategies are for passenger vehicles and residential buildings, which means it is not comprehensive. In addition, the Grubler et al. (2018) paper also did a global assessment of ME strategies (all, not just cars and houses) under changing socioeconomic and energy supply contexts.

>> We have pointed out the limitations of the approach of Grubler et al. above in the discussion of novelty. In response to this comment, we have modified the sentence to:

L360: “This study provides the first detailed assessment of ME strategies in two major end-use sectors with a global scope and in a changing socioeconomic and energy-supply context.”

L254: “While service, product, and material cycle modeling is largely absent from integrated assessment models,” Again, some mention of the Grubler et al. (2018) paper is surely required here, as they have integrated the ME assessment into the IAM modelling (albeit exogenously).

>> We partly agree and now cite Grubler here for service and product modelling, but not for ME/material cycle modelling, as the Grubler representation of ME is rather stylistic (as described above).

We rephrased and extended, now L369: “Material-cycle modeling is largely absent from integrated assessment models, which are the work-horses of global climate-mitigation

assessment, and assessments such as the one presented here can be soft-linked to and possibly integrated into such models similarly to how land-use modeling has recently been integrated. Soft-linking would help establish the stock-flow-service nexus,²⁷ ME strategies and material-cycle and resource constraints in climate-mitigation scenarios,³⁹ and integration would allow for including ME into optimization routines. Better integration into large-scale assessments would also allow us to study the global economic implications of ambitious ME.”

>Materials and Methods

This section describes several methods and databases, including RECC, stock-flow-service-nexus, SSP, WEO, ETP, MESSAGE IAM, etc. Alongside this, it refers to data and analyses from numerous other papers. Furthermore, Table 1 gives only a generic description of the methodological approach. I afraid this reads more like a literature review of models, than a method section of what you did for this paper. And it is unclear whether what actually modelling work was completed for this specific paper (as opposed to the many others listed by the authors). I suggest this needs a complete rethink to clearly delineate between the models and data you use which has already been published, and the additional research completed for this paper which is not already published. Without this, it is difficult to ascertain the novelty of this paper.

>> We appreciate the comment. We have reorganized large parts of the introduction and the entire methods section to first describe the method in terms of the RECC model, and then the specific parameters and scope that apply to the present manuscript (see new Table 2). Given the questions raised by the reviewer of the relationship of our work compared to that of Grubler et al., we are now more explicit in the text (and very detailed in the Supplement 1, section 1) about describing the material flow analysis methods (which are lacking in Grubler). As explained in our response to the concern about overlap with previous work, we have described the global scope and coverage of material stocks in the present study.

Introduction, L139: “Our analysis covers the resource and GHG impact of ME in these two sectors, covering the entire world comprised of 20 countries/regions, grouped into the Global North (OECD, former USSR countries, China) and the Global South (low and medium-income countries in Asia, Africa, and the Americas). It represents a significant advancement over previous work¹¹ as it covers the entire world and all emerging economies, in particular, and because the scope of the research questions was expanded to quantify the impacts of ME on material stocks and production as well as on the vehicle and building stocks and changes in stocks.”

The new Table 2 in the methods section now reports a summary of the relevant (and new) data, and all figures (1-4) report entirely new results at the global, Global North, and Global South levels.

Reviewer 2

Reviewer #2 (Remarks to the Author):

In this manuscript, the authors present a comprehensive dynamic Material Flow Analysis, with GHG emissions link, for passenger vehicles and residential buildings from 2016 to 2060. They do this to compare the effect of GHG emissions reduction potential of Material Efficiency strategies with climate policy strategies. Analyzing co-benefits between circular economy and climate neutrality targets is highly relevant research. The extensive model ODYM-RECC, utilizing the newly developed ODYM framework, quantifying results for 20 world regions and considering material and energy flows in 16 processes including a use-phase of various car types and building types (“archetypes”) is definitely new in peer-reviewed research. Parts of the results and model have been published in the International Resource Panel report on material efficiency and similar publications, or as partial preprint documents. The results are interesting, because they highlight the existence of anticipated co-benefits, the need for material efficiency measures to reach anything close to climate neutrality and the robustness of these results based on different scenarios including Shared Socioeconomic Pathways. In particular, the transparency and accessibility of this research project is remarkable.

Major comments:

A) Time frame: I’m missing a good justification for choosing 2060 as last year in the dMFA and 2050, in contrast, as year for evaluation, e.g. in Fig. 2. It is a bit confusing that the abstract talks about GHG emissions “until 2060”, but the results section shows annual and cumulative results for 2050. A reason for different reference year, or extending the model calculations beyond the main year of interest (2050) is not given. You should consider either harmonizing, or better justifying the choice of time frames in your article.

>> We agree and now use 2050 as time horizon in the paper. The SI reports results until 2060. The text and table 1 (model resolution) now state “Time dimension: 2016–2060 in steps of 1 year, results are reported for/by 2050.”.

B) Archetype description and results display: I see a potential mismatch between vehicles and buildings in your model, at least in the way you show the intermediate results for vehicle and building stock in the Supporting Information. While the vehicles are shown disaggregated for fuel type, the buildings are shown disaggregated by energy efficiency standard and number of households in the building. If I understood your model correctly, then these are not the “archetypes” of vehicles and buildings in your model, but they are energy efficiency measures. I think it would be more intuitive if the results for the building stock were shown in terms of heating fuel type and cooling device, rather than building size, because this would be the equivalent to vehicle fuel type. And heat pumps, for example, are only mentioned very briefly in l. 159 in the main manuscript. The equivalent to building size would be car type (micro car to light truck), which however is part of your

archetypes. Therefore, I see a need to clarify which parameters go into archetypes, which parameters are energy efficiency measures and which parameters are material efficiency measures.

>> **[Added to supplementary material 1, section 3.4]** “The building and vehicle archetypes differ indeed by their characteristics and are being modeled very differently. The energy source is a discrete property of vehicles and (normally) cannot be changed during its lifetime, so it is considered a defining parameter. In the case of buildings, however, energy supply can be considered independent of a building as it is common for a building to change its thermal energy generator during its lifetime.

This is the reason why we decided to use different approaches when modelling vehicle and building archetypes.

The building archetypes have the following discrete dimensions that are being modeled endogenously by the archetype model (BuildME):

- occupation (3 types: multi family, single-family, residential tower)
- energy standard (4 types: non-standard, standard, efficient, zero-energy)
- resource efficiency strategy (4 types: none, light-weighted, material substitution, combined)
- +1 informal type that is considered constant.

In addition to that, the archetypes are simulated for each climate zone individually. Building energy supply is considered exogenously and modeled by the main stock model (ODYM).

The vehicle archetype is mostly characterized by its drive-train technology and has 6 different types, therefore determining the energy demand endogenously.”

C) Current trajectory of policies: You model three different pathways for society (LED/SSP1/SSP2). Your model starts in 2016, but we are already in late 2020. Could you mention, anywhere, which path we are currently following? I'm most familiar with GHG emissions from passenger vehicles and from what I see in Fig. S-7, I would say we are much closer to SSP1 and very far away from LED scenario. If you have data on this, this could be worth a critical note in your manuscript.

>> Very interesting and relevant question! We added the following to the supplementary material 1, section 4: “We have not yet investigated in detail the question on which GHG emissions trajectory the two studied sectors are currently on. An update will be necessary in the coming 2-3 years to trace the actual development, assess the effect of existing policy, and adapt the scenario assumptions. Between 2015 (base year of the RECC model) and 2020, urban expansion continued and related GHG emissions rose in most world regions, and the actual development roughly followed the SSP1 and SSP2 trajectories. The LED scenario is a stark deviation from current growth patterns and its service levels would only be attainable in a world with very strong climate policy, combined with widespread lifestyle changes. The COVID 19 pandemic will have an impact on both growth (economic development) and lifestyle in the different regions, which will be visible in the emissions patterns together with the effect of climate policy.”

D) Public transportation: Looking at Fig. SI4-1 (Drivers and in-use stock parameters, USA), I'm curious whether the assumed difference in passenger kilometres (which is essentially the functional unit) comes from a shift in modal split, or from a reduced

mobility demand. If the first one is the case, then there should be at least a discussion whether infrastructure expansion for public transportation might lead to trade-offs in terms of material efficiency and GHG requirements. At this moment, public transportation seems to be completely outside of your model and this not discussed anywhere in the manuscript. In the end, the most effective ride-sharing option is public transportation, isn't it?

>> We thank the reviewer for this thoughtful comment. In this work, the development of passenger km by scenario is solely a function of occupancy rate and vehicle km. Public transport is currently not modelled, and therefore changes to modal split are not considered. As such, availability and expansion of public transport does not influence passenger km or vehicle km. Future versions of ODYM will likely include representations of public transport, therefore the trade-off of lower vehicle km versus higher material requirements needs to be considered, as the reviewer correctly pointed out. Since public transport is not considered we did not address this trade-off in the manuscript.

Minor comments to the manuscript:

1) L. 56ff: In the last sentences of the abstract, the order of passenger vehicles and residential building is reversed, should be the same.

>> Fixed.

2) L. 60: Last sentence should end with "for passenger cars" instead of "for residential buildings".

>> Fixed.

3) L. 67: What does "rate-limited" mean in this context? Expression is not clear to me.

>> Rephrased into "The decarbonization of industry and material production, in particular requires technological and organizational change and large investments into new energy infrastructure and factories" (L72).

4) L. 70: Typo: "of" instead of "if".

>> Fixed.

5) L. 73: Why is this a competition? Can't we do both decarbonizing material production and low-carbon energy provision?

>> Eventually, yes. But for the coming decades, low-carbon energy will be a scarce resource and demand for it will be much higher than supply.

6) L. 115: Would be helpful to name SSP1 and SSP2 here. In Fig. 4.1 of the Documentation, this is "Low Challenges and Intermediate Challenges".

>> Agreed and we added to now L148: "representing low and intermediate socioeconomic challenges related to climate change adaptation and mitigation, respectively."

7) L. 154: You introduced the terminology of Global North and Global South, so it's better to stick to that other than using the term "developing countries".

>> Agreed, also because we want to avoid the term 'developing countries'.

8) L. 170: Isn't the additional emissions savings of 1-2 Gt/yr too big? Total reductions from resource efficiency are at max 2 Gt/yr in Fig. 1.

>> It's a different account. There is already a lot of wood being used in the no-ME scenarios, and Fig. 1 (now Fig. 2) only shows the delta. We make this clear by adding to L248: " , depending on how much of it is used."

9) L. 182-189: The whole paragraph uses unnecessarily complicated sentences and the meaning overall is not clear. Try to simplify your expressions. If I understood your model correctly, the baseline for 2050 annual GHG emissions varies between 1.8 Gt (LED+RCP2.6) and 15 Gt (SSP2+NoPol). From this baseline, the ME-induced

reductions for 2050 annual GHG emissions cut between 1.1 Gt (SSP2+NoPol) and 3.4 Gt (LED+RCP2.6). This means the lower the baseline, the higher the relative additional emission reductions from ME (61% for LED+RCP2.6 and 22% for SSP2+NoPol). And the relative GHG reductions are bigger for 2050 annual emissions than for cumulative emissions until 2050, because currently the baseline is still high. Is that correct?

>> Exactly! To make this clearer, we rephrased the whole passage was rephrased into L192: “Global GHG emission savings of ME: The different ME strategies can reduce cumulative global GHG emissions of the period 2016-2050 by 32-77 Gt (13%–18% of the total), depending on socioeconomic development and climate policy (Fig. 1, top row, Table 2 for scenario settings). All examined strategies show a visible contribution (numerical values reported in supplement 3). For the LED scenario, where in-use stocks are already used very intensively (low floor space per capita), material substitution, re-use and longer use are the ME strategies with the largest GHG reduction potential. For SSP1 and SSP2, more intense building use and material substitution show the largest contribution, followed by downsizing, re-use, and longer use. The ME strategy car-sharing shows much larger contributions in the 2°C policy mix. The reason for that is that this scenario has a higher share of electric vehicles, which are introduced faster, because car-sharing reduces the vehicle fleet size but increases the average annual kilometrage, thus shortening vehicle lifetime, which increases the turnover of the fleet.

Once fully implemented, ME strategies can lead to large reductions of annual global GHG emissions. In 2050, annual savings can be between 22% and 61%, depending on ME stringency, energy-sector decarbonization, and anticipated growth in services (Fig. 1, bottom row).”

10) Fig. 2: This figure should be improved: 11800 Mt can not be read well. It is not clear enough that the first number applies only to the brown Bottom line.

Capitalization in the legend is inconsistent. The ME measures listed in the legend are not described, or even mentioned in the text so far.

>> This figure was carefully revised, including the changes that you are asking for here. It appears now as fig. 1 in the paper, please check. We also added a new table 2 in the methods section where all ME strategies are defined and their main quantitative potentials are shown.

11) L. 334: Typo: “product-“ instead of “product”.

>> Fixed.

12) L. 352: I guess the comma should read “or” instead. It is either Excel sheets or scripts. Or, alternatively, did you miss writing about a third option?

>> Rephrased to ‘both ... and ...’.

13) Table 1: Does “other” refer to wood and plastics? Why are they not modelled as wood and plastic separately?

>> This is the list of chemical elements. ‘Other’ denotes all elements for which there is no separate mass balance, because they are not within the scope of the analysis.

14) L. 411: Either “and “ or “to” are not needed here.

>> Fixed.

15) L. 555f: “A.N.” appears twice in the list for archetype and scenario formulations.

>> Fixed.

16) L. 479: Ref. Pauliuk et al. 2020 volume, issue, pages need update.

>> Fixed.

Minor comments to the Supplementary Material 1:

17) Figure numbers: There is a mismatch between numbers given until Fig. SI4 (no hyphen) and starting with SI-5 (with hyphen). In section 4.6 it's all of a sudden SI1-15 (and so on) instead of SI-15. Please align the numbering and formatting.

>> Fixed!

18) Figs. SI-5-1 to SI-6-20: Changes between top row and bottom row would be much easier to grasp if y-axis limits were chosen the same. Also it would be good if title or axis labels, or legend, mentioned the difference in "no ME" and "full ME" spectrum. The reader should not have to search for this important information in the figure caption.

>> Important feedback! Will include this in the next model revision.

19) Fig. SI-5-4: Why is there a significant share of HEV in all European regions, but not in Germany? Seems inconsistent.

>> The drive technology shares for Germany are taken from a different source, see parameter documentation and DOI 10.1111/jiec.13091.

20) Figs. SI-5-1 to SI-6-20: I don't get the order of countries. It's alphabetical, not by size of the country and not by world region either. With 20 regions, this should follow some intuitive logic and should be consistent throughout the manuscript and supplementary.

>> The order here follows the internal order of 20 regions in the model config file. Clearly, this is not intuitive, but given the large number of regions, we think it is better anyway to start with table SI1-6 and search the pdf for the plots with the region of interest.

21) Figs. SI-7, SI-8: Please split this into three figures each, so that caption and figure are always on the same page.

>> These plots are actually parts of a single figure, they just don't fit onto a single page.

22) Figs. SI-7, SI-8: Please don't just write pav and reb in the figure titles, write "Passenger vehicles" and "Residential buildings" instead.

>> Added to headline and caption.

23) Figs. SI-7, SI-8: The order of countries in the 5x5 subplots changes between the figures. As mentioned above, the order of countries should always be consistent and follow some logic.

>> For these figures, we also have the challenge that the y axis should be comparable or at least grouped across each line, to allow for some easy comparison. That is what the country order differs between the vehicle and the buildings plot.

24) Fig. SI-9: Why does such a figure exist only for passenger vehicle sector? Is the buildings figure missing?

>> It was only created for passenger vehicles, by the vehicles team. Because of its flawed axis, the figure was removed from the SI, see next comment.

25) Fig. SI-9: Please don't use bar graphs with logarithmic scale axis. In particular not stacked bar graphs. It can lead to the case that smaller values get more inked area, which is counter-intuitive and therefore should be avoided in any graphical display of data.

>> Fully agree! I overlooked that and this figure was removed from the SI.

26) Section 4.3: Don't use pav and reb abbreviations in section headings.

>> Fixed.

27) Fig. SI-11: Again, the Figure caption should not be the only place to distinguish left and right plots. They have exactly the same titles, labels and legends. Please add any information on baseline and RCP2.6 to these graphs.

>> Fixed.

28) Fig. SI-11: I'm confused by the passenger vehicle graphs because of (most likely) negative values in a stacked bar graph. The dark colour does not appear in the legend and the order of elements seems to change.

>> Exactly. The disclaimer "Colors may deviate from legend colors due to overlap of RES wedges" was added.

29) Section 4.6: Is this primary energy or final energy consumption?

>> It's final energy, which we added to the section heading and the figure captions.

30) Fig. SI1-23: The exponents (1e8 and 1e7) are somewhat distracting and overlapping with the other graphs.

>> That seems to be a pyplot but that I haven't been able to fix.

31) Fig. SI1-24: Units are not clear. Passenger-kilometer per GHG kilometer/tonne? What is a GHG kilometer?

>> A comma is missing here and was added. Caption is "passenger km per GHG, km/t".

32) Ref. Reyna et al. 2014: Volume, Issue, Pages wrong

33) Ref. Riahi et al. 2017: Volume, Issue, Pages missing

>> Fixed.

Minor comments to the ODYM-RECC Documentation:

34) Box 1: "System boundary" should not be capitalized.

>> Fixed.

35) Table 1.1: CaS and RiS should only apply to vehicles and MIU should only apply to buildings, if I'm not mistaken.

>> Fixed.

36) Fig. 3.4: Why are % split in pass. Vehicles, trains, bus, etc. and % split in to single and multi-family houses, apartment blocks "currently not implemented"? What does this mean to the model? Don't you show exactly such a building type split in the Supplementary Material 1?

>> That is a mistake indeed, which we corrected. For transport, we currently do not have a modal split, only passenger vehicles. For buildings, the split is present, and we removed this remark on the building side.

37) Table 3.1: Maybe just a comment for the future: What are you going to do if your model, at any point in time, exceeds 52 indices? Is there any possibility to extend index letters beyond this point? (Probably needs to be addressed in ODYM, not ODYM-RECC).

>> That is a good point! The main user of the index letters is the Einstein sum function of the Python programming language's numpy library, which only accepts latin alphabet letters for index labels. The best way out would probably be to extend the index table to e.g., Greek letters or Chinese characters and then translate them to latin letters for each individual calculation whenever needed.

38) P. 26: See above, what is "other" elements?7

>> 'Other' refers to the sum of all elements not explicitly listed, for mass balance. We added that description to the list. For example, when Fe and Ni are the only two elements explicitly

traced, C, Cr, Cu, etc. would be part of 'Other', and the mass of a steel flow would be the sum of the masses of Fe, Ni, and 'Other'.

39) Fig. 4.2: Because you are using SSP1 and SSP2 in the model, why do you show SSP2 and SSP5 here?

Fig. 4.2.: PPP GDP per capita in SSP2 (left) and SSP5 (right), by region. Vertical dashed line: 2018, horizontal dashed line: 40000 US\$2005 per capita. Negative values are contained in SSP database. Own plot.

>> Just as an example. We now added this statement to the figure caption: "Note that in the current model setup, only the SSP1 and SSP2 scenarios/storylines are included. The ODYM-RECC model does NOT use GDP as a model driver/parameter."

- 40) Ref. Bürger et al. 2018: Published where and how?
- 41) Ref. Deetmann et al. 2019: Volume and Issue missing
- 42) Ref. Deetmann et al. 2018: Issue and Pages missing
- 43) Ref. Elshkaki et al. 2018: Volume, Issue, Pages missing
- 44) Ref. Haberl. Et al. 2017: Pages missing
- 45) Ref. Hertwich et al. 2015a/b looks like the same publication.
- 46) Ref. Laner et al. 2014: Volume, Issue, Pages wrong
- 47) Ref. Nakamura et al. 2017: Please refer to the ES&T publication, not a preliminary draft.
- 48) Ref. Riahi et al. 2017a/b looks like the same article
- 49) Ref. van der Voet et al. 2018: Volume, Issue, Pages missing
- 50) Ref. Winning et al. 2017: Volume, issue, Pages missing

>> Thanks for checking these details! We corrected the reference list accordingly.

Overall, I expect that the authors will be able to address these issues. I recommend inviting the authors to revise their manuscript.

Christoph Helbig

Reviewer #3

Reviewer #3 (Remarks to the Author):

Authors present a comprehensive global assessment of Material Efficiency strategies for climate mitigation. This is a timely contribution, given the high policy priority

around resource efficiency and climate change. One of the unique aspect of this study remains the methodological combination of design parameters and product life cycles with operational and embodied climate impacts for residential buildings and passenger vehicles. From practice point of view, all possibilities of major material efficiency strategies have been included in the assessments. There is an adequate justification for specific materials and scenarios.

From a methodological perspective, I have a few comments for authors:
One of the major challenge in this research, as I see, is associated with the huge range of expected benefits and a significant uncertainty in the estimates. For example, given the global nature of coverage- one may argue that it may be possible to create limited archetypes for passenger vehicles but the material intensity factors and operational energy for residential buildings would take more then the archetypes one can ideally agree upon. How do authors see these issues influencing the results?

>> **[Added to supplementary material 1, section 3.4]** “The beauty of the bottom-up archetype modeling approach is that it can be used to explain very large and heterogeneous product systems. Other modeling approaches, such as top-down or statistical models, have a much lower degree of freedom when it comes to applying engineering strategies (e.g. effect of increased insulation material in different climates). As the reviewer mentions, the trick for the bottom-up approach is to strike a good balance between true representation of the system and model complexity.

In our case we decided to choose the most common technological systems today and what we consider the most hopeful technologies in the intermediate future.

In case of the building archetype model we are somewhat at the upper limit of model complexity. The archetypes represent 3 occupations, 4 energy standards, 4 Resource Efficiency strategies, and approx. 30 climate zones. This results in ca. 1400 manifestations. As we are using complex thermal simulation tools, model runtime is already substantial. Yet we decided to make this tradeoff as we deemed all of the aforementioned parameters as important enough to account for them explicitly.

To summarize, for the building archetypes, we take the region-specific climate into account but no behavioral aspects, which is a limitation that we now describe in more detail in the manuscript (see quotation below). We could simulate different material choice scenarios on top of the concrete-wood-substitution that we already model. But this research topic/question would warrant a different paper, ideally, a detailed country-level case study, which the RECC model can also be used for.”

I can see that a very limited number of climate mitigation assessments of material efficiency strategies have informed the global estimates of potential emission savings in this study. Do authors find these sources sufficient to generalize impact values for all countries?

>> We are not sure what the reviewer means. Our study provides an estimate of potential emissions savings based on modeling and upscaling of region-independent vehicle archetypes and region-(climate)-dependent building archetypes as described. Throughout

this work, we had to make assumptions where there was insufficient data, e.g. about the breakdown of the existing building stock into different types and energy standards in much of the world. Better empirical information on the starting point would certainly be desirable, but our approach has been designed to accommodate existing uncertainties.

Even though technology and energy aspects over product life cycles have been combined in this study, Behavioral aspects of firms and users have been missed out of the equation. User behavior is arguably one the biggest influence in the material efficiency adoption at scale. Choice of sharing a vehicle or reusing a window or building component is highly consumer dependent. Would it be wise to have scenarios influenced by the consumer acceptance?

>> The way we deal with behavior is by assuming different adoption rates in different scenarios. The reviewer is completely correct that we do not explicitly model behavior, but rather just make assumptions and see how it plays out. A different type of research would be needed to address those behavioral aspects. In response to this comment, we now discuss the possible extension of the model with costs and the introduction of an economic logic, which would allow to model cost-minimizing behavior. There are, of course, other logics that are also important for firm and consumer behavior, especially with respect to cars and residential buildings, but since we have only very limited space to address this issue, we refrain from going into these aspects.

In the discussion section of the manuscript, we now write on L382:

“The RECC results represent estimates of the technical potential of ME. To estimate the feasible potential of ME under different business models and policy scenarios, material production and recycling costs need to be included, among others. Adding the cost layer to the material cycles would allow for circular economy business model simulation for ME41 and the estimation of employment impacts.⁴² Combined with macro-economic modelling, cost information would enable us to quantify rebound effects⁴³ due to lower material prices from under-utilized primary production assets and increased availability of (lower quality) recycled material.⁴⁴ Including costs would facilitate the simulation of policies to mitigate material efficiency rebounds, such as eco-design standards, cap and trade systems for recourses, or raw material extraction taxes.”

Some minor comments:

Line 60- passenger vehicles instead of residential buildings?

>> Indeed. Fixed.

Line 70- 'now 23% if' Is this statement still true accounting for the expected decline in GHG emissions in 2020? It would be helpful to mention the year or remove 'now'.

>> Changed into “represent about 23%”. (L75).

Line 78- Supply-side mitigation seems ambiguous term and the cost reduction may not necessarily be true given the complexity of trade-offs.

>> Agree that this phrase needs more explanation. Because we stress the fact that material efficiency can lower the demand for supply-side mitigation (low carbon energy, bioenergy-CCS, CCS, ...), and thus also costs, at several places, we delete this phrase here.

Line 80- 'Reducing material use' seems ambiguous. Would it be more appropriate to say that these measures are to reduce material demand and increase/maximize the material use?

>> We agree and rephrase into 'reduce material demand'. (now L85).

Line 86- re-use of scrap seems to included in both fabrication efficiency and waste management- can authors clarify this aspect?

>> Correct! One is the re-use of fabrication scrap for other purposes, like manufacturing small parts. The other one is the re-use of end-of-life products and components. We clarify on L118: "reuse of fabrication scrap".

Line 177-180 In assessing the annual and cumulative GHG emissions savings, what is the expected scale of adopted ME measures? Full ME adoption- does it include all ME strategies adopted by complete existing and upcoming stocks of buildings and vehicles?

>> We agree that more context and detail needs to be provided here, and added an entirely new table (Table 2) to the methods section below to indicate the implemented ME strategy potentials.

REVIEWER COMMENTS

Reviewer #1 (Remarks to the Author):

This paper presents an assessment of the potential GHG emissions reductions from applying material efficiency saving in residential buildings and cars. The paper is well written and includes an extensive Supporting Information document.

The authors have responded to each of the reviewers' comments in detail, and made significant changes to the paper in response. These edits make for a much better read. I am now happy that the paper is in an acceptable form to publish.

Reviewer #2 (Remarks to the Author):

The authors have made substantial efforts to address the reviewer comments. Some comments remain from my side, some of which are caused by the authors' adaptations to other reviewers' comments.

Major comments:

A) Primary vs secondary production of copper: In response to one reviewer, you added a sentence on the dates when secondary production exceeds primary production, equivalent to the recycled content exceeding 50%. For copper, you say the model puts this before the year 2020. Which would mean this already happened. To my knowledge, the RCR for copper has always been and is still below 40%. Therefore, this needs some attention. It could be an artefact in your data. If it persists, then this requires at least a comment in the manuscript.

Minor comments:

- 1) l. 105f: The sentence on Van Ruijven et al. is not well implemented into the paragraph. If it is just one example, please add "For example," or similar. If you want to add another example, please do that. If no change is made, the reference of the following sentence starting with "These scenarios" is unclear.
- 2) Inconsistent use of CO₂e vs CO₂-eq. in text and figures.
- 3) I'm afraid, Fig. 3 has suffered from adding the reversed order of EE and ME measures. It's not intuitive anymore. I think the focus of the figure is now not clear anymore. What is the key message? For me, it's counter-intuitive to have one stacked bar chart (as a summary of a waterfall diagram) on the left of the "no climate policy" baseline and one on the right. Would it help to have the alternative series of policy implementations shown only in a variant of the figure in the Supplementary Materials? Maybe, but I'm not sure what's the best solution. As it is now, I know that the graph is hard to interpret without detailed text knowledge.
- 4) Fig. 4: Description says data shown is 2016-2050. However, the x-axis seems to go beyond 2050. I'd guess that the update from 2060 to now only until 2050 has not been adequately updated.

Christoph Helbig

Reviewer #3 (Remarks to the Author):

Thank you for appropriately addressing my comments and suggestions. Manuscript makes significant academic advancement and an important contribution towards material efficiency momentum across the globe. Congratulations.

I am satisfied with the revision and recommend it for publication.

Reviewer Comments

Reviewer #1 (Remarks to the Author):

This paper presents an assessment of the potential GHG emissions reductions from applying material efficiency saving in residential buildings and cars. The paper is well written and includes an extensive Supporting Information document.

The authors have responded to each of the reviewers' comments in detail, and made significant changes to the paper in response. These edits make for a much better read. I am now happy that the paper is in an acceptable form to publish.

>> We thank you for reading and assessing our response letter and the revised document!

Reviewer #2 (Remarks to the Author):

The authors have made substantial efforts to address the reviewer comments. Some comments remain from my side, some of which are caused by the authors' adaptations to other reviewers' comments.

Major comments:

A) Primary vs secondary production of copper: In response to one reviewer, you added a sentence on the dates when secondary production exceeds primary production, equivalent to the recycled content exceeding 50%. For copper, you say the model puts this before the year 2020. Which would mean this already happened. To my knowledge, the RCR for copper has always been and is still below 40%. Therefore, this needs some attention. It could be an artefact in your data. If it persists, then this requires at least a comment in the manuscript.

>> That is a very important observation and we added the following clarification/discussion on L321f:

“The material production volumes (Fig. 4) only include the demand and scrap supply of the two sectors studied, and the ratio between primary and secondary production reflects the sector-specific material stock dynamics and not the global total for the individual metals. Copper is an interesting example here, as its global average recycled content is below 40%, mainly due to large losses in electronics,^{39,40} but for vehicles and buildings, scrap recovery rates are high and the recycled content in the material supply for these two sectors can be 60% and higher.”

Minor comments:

1) L. 105f: The sentence on Van Ruijven et al. is not well implemented into the paragraph. If it is just one example, please add “For example,” or similar. If you want to add another example, please do that. If no change is made, the reference of the following sentence starting with “These scenarios” is unclear.

>> Revised/rephrased, the passage now reads:

“Industrial ecology research has provided a number of scenario analyses for the future demand and supply of specific materials and metals.^{21–23} In the integrated assessment

modelling community, van Ruijven et al.²⁴ developed a GDP-driven high resolution scenario model for steel and cement demand and production. However, these material demand and supply scenarios have not been linked to service provision and do not include a detailed depiction of ME.”

2) Inconsistent use of CO₂e vs CO₂-eq. in text and figures.

>> Thanks for spotting this inconsistency! It should be CO₂-eq all throughout and this has now changed.

3) I'm afraid, Fig. 3 has suffered from adding the reversed order of EE and ME measures. It's not intuitive anymore. I think the focus of the figure is now not clear anymore. What is the key message? For me, it's counter-intuitive to have one stacked bar chart (as a summary of a waterfall diagram) on the left of the “no climate policy” baseline and one on the right. Would it help to have the alternative series of policy implementations shown only in a variant of the figure in the Supplementary Materials? Maybe, but I'm not sure what's the best solution. As it is now, I know that the graph is hard to interpret without detailed text knowledge.

>> We added the alternative sequencing to respond to the clear request by reviewer 3. For this revision, we redrew the figure to remove the middle bar and also indicate that the bar segments represent reductions. Please see the new figure in the manuscript, Next to simplifying and clarifying the figure, we think that your concern can be best addressed with a rephrased figure caption, where the sequence and meaning of the different bars/columns is introduced early on:

“Fig. 3: Breakdown of total emission savings from baseline with no new climate policy (black horizontal line on top of bars) into end-use energy efficiency, energy supply, industrial and demand-side material efficiency, for passenger vehicles and residential buildings combined, at the global level (a), the Global North (b), and the Global South (c). For the left bar in each scenario, ME was implemented first, before adding energy efficiency and low carbon energy supply. For the right bar, ME was applied in addition to energy efficiency and low carbon energy supply. The two red-colored segments cover the ten ME strategies. Industrial ME includes recovery ratios for recycling, fabrication yield and scrap diversion, re-use, and material choice. Demand-side ME includes product light-weighting/downsizing, lifetime extension, car sharing, ride sharing, and more intense use of buildings.”

4) Fig. 4: Description says data shown is 2016-2050. However, the x-axis seems to go beyond 2050. I'd guess that the update from 2060 to now only until 2050 has not been adequately updated.

>> We now provide a revised version of Fig. 4 that shows values up to 2050 only.

Christoph Helbig

Reviewer #3 (Remarks to the Author):

Thank you for appropriately addressing my comments and suggestions. Manuscript makes

significant academic advancement and an important contribution towards material efficiency momentum across the globe. Congratulations.

I am satisfied with the revision and recommend it for publication.

>> We thank you for reading and assessing our response letter and the revised document!

REVIEWERS' COMMENTS

Reviewer #2 (Remarks to the Author):

In my view, the authors have successfully addressed all remaining comments from the previous round of reviews.